# Aero-servo simulations of an airborne wind energy system using geometry-resolved computational fluid dynamics

Niels Pynaert<sup>1</sup>, Thomas Haas<sup>3</sup>, Jolan Wauters<sup>1,2</sup>, Guillaume Crevecoeur<sup>1,2</sup>, and Joris Degroote<sup>1,2</sup>

**Correspondence:** Niels Pynaert (Niels.Pynaert@UGent.be)

Abstract. Airborne wind energy (AWE) is an innovative and promising technology for harnessing wind energy, often achieved through the use of tethered aircraft flying in crosswind patterns. A comprehensive understanding of the unsteady interactions between the wind and the aircraft is required for developing efficient, reliable, and safe AWE systems. High-fidelity simulation tools are essential for accurately predicting these interactions. To provide meaningful insights into crosswind flight maneuvers, they should incorporate the coupled nature of aerodynamics, dynamics, and control systems. Moreover, local aerodynamic phenomena, such as flow separation, play a significant role in the overall performance of the system and must be represented accurately. Capturing these phenomena requires resolving the complete geometry of the aircraft. Therefore, this work presents a geometry-resolved computational fluid dynamics framework of an AWE system, encompassing all lifting surfaces and integrating movable control surfaces, referred to as the virtual wind environment (VWE). Unlike existing models that only consider linear combinations of individual aerodynamic effects, the VWE addresses the challenge of combining the relevant aerodynamic interactions specific to crosswind flight motion. This VWE is coupled to the dynamics and control framework of an AWE system, enabling geometry-resolved aero-servo simulations. We demonstrate the novel simulation framework by tracking a pre-optimized 1-loop power cycle in the VWE coupled to model predictive control, achieving 96% of the reference power.

#### 15 1 Introduction

Airborne wind energy (AWE) is an innovative and promising technology to harness wind energy and convert it into electricity, for example using tethered aircraft flying in crosswind patterns. A key advantage of AWE systems is their potential to operate at higher altitudes than conventional wind turbines, where the wind is stronger and more consistent (Diehl, 2014). Additionally, AWE systems require significantly less material for the same power generation, as the need for the tower and blade material near the axis of rotation is eliminated. Two main operation modes for energy conversion are currently pursued within emerging companies and academia: on-board generation (fly-gen) and on-ground generation (ground-gen). Furthermore, both soft kites and fixed-wings are utilized as tethered aircraft (Cherubini et al., 2015; Vermillion et al., 2021). This work focuses on the mod-

<sup>&</sup>lt;sup>1</sup>Department of Electromechanical, Systems and Metal Engineering, Ghent University, Sint-Pietersnieuwstraat 41, Gent, 9000, Belgium

<sup>&</sup>lt;sup>2</sup>MIRO core lab, Flanders Make @ UGent, Sint-Pietersnieuwstraat 41, Gent, 9000, Belgium

<sup>&</sup>lt;sup>3</sup>Thermo and Fluid Dynamics (FLOW), Faculty of Engineering, Vrije Universiteit Brussel (VUB), Pleinlaan 2, Brussels, 1050, Belgium

eling and simulation of fixed-wing ground-gen systems with a fixed ground station, which operate using a so-called pumping cycle. This cycle consists of a reel-out phase, during which the aircraft pulls on the tether and energy is extracted, and a reel-in phase, during which the tether is rewound, consuming a portion of the energy converted during the reel-out phase.

AWE systems experience a dynamic interaction between the aircraft, the atmosphere, and the controller. The presence of non-ideal wind conditions can induce unsteady aerodynamic phenomena, and its prediction remains an open challenge (Vermillion et al., 2021). Reliable control systems are crucial for steering the aircraft and operating the system safely. The fixed-wing system employs control surfaces to steer the aircraft, similar to conventional aircraft. Accurate prediction of aerodynamic forces and moments while deflecting the control surfaces and their impact on the system dynamics is crucial for designing safe control systems. Moreover, precisely tracking the intended flight path is important for the system's performance. The varying flow velocity encountered during crosswind flight maneuvers also influences local aerodynamic phenomena, such as flow separation. Because these phenomena significantly affect the overall performance of the system, they must be accurately represented in simulations, necessitating resolving the complete geometry of the aircraft in the simulations.

To study local aerodynamic phenomena, several studies have investigated various designs of fixed-wing AWE systems using computational fluid dynamics (CFD). Eijkelhof et al. (2023) developed an aerodynamic toolchain for the design analysis of AWE systems with a box-shaped wing using steady Reynolds-averaged Navier – Stokes (RANS) simulations. Vimalakanthan et al. (2018) conducted RANS simulations of a double fuselage aircraft that included control surfaces. However, both studies focus on a steady horizontal flight, neglecting crosswind motion and setting the control surfaces in a fixed position. Kheiri et al. (2022) examined the wake flow of an AWE system using unsteady RANS for both the aerodynamics and wind simulation, assuming circular flight motion and considering only the main wing, while omitting control surfaces. Castro-Fernández et al. (2021) incorporated more complex motion in their aerodynamic simulations by prescribing crosswind flight motion to a panel representation of the wing, without control surfaces, using the vortex lattice method (VLM). In the work of Fasel et al. (2019), a fully coupled aero-servo-elastic framework was used with a 3D panel method representation of the wing, focusing on optimization studies for morphing wings. Similarly, the work of Haas et al. (2022) and Crismer et al. (2024) presented an actuator line representation of the wing within a Large Eddy Simulation (LES) framework coupled to 3 degrees of freedom (DOF) and 6-DOF aircraft representations, respectively. However, in Haas et al. (2022), only the main wing is considered and Crismer et al. (2024) uses an analytical model for control surface deflections. In both cases, the local aerodynamics are calculated from pre-computed steady airfoil polars, thus neglecting unsteady effects. In conclusion, none of the previously conducted studies on AWE have combined the aircraft's motion in geometry-resolved CFD simulations coupled to the system dynamics and control. The current state-of-the-art AWE simulations often rely on tracking pre-defined flight trajectories using modeling and optimal control tools, such as AWEbox (De Schutter et al., 2023). These tools typically employ aerodynamic models based on pre-computed coefficients or fast analytical approaches, such as the stability-derivative-based model proposed by Malz et al. (2019). While these aerodynamic models are computationally efficient, they exhibit notable limitations. Specifically, they fail to account for various aerodynamic effects, including unsteady phenomena and interactions, such as the influence of the rotational speed on the effectiveness of the ailerons, which can only be captured through flight testing or high-fidelity CFD. Such effects can lead

to violations of the constraints defined during trajectory optimization, potentially resulting in degraded performance or even structural failure. Given the high cost of flight testing, accurate CFD tools are essential to predict and mitigate these effects during the design and planning stages.

This work presents a geometry-resolved CFD framework coupled to a controller, which is described in the next paragraph. The CFD framework integrates both the motion of the AWE system and the movement of control surfaces, including ailerons, elevators, and rudders. We refer to this comprehensive CFD framework as the virtual wind environment (VWE) and it uses the Chimera/overset technique, previously applied to simulate control surface deflections for general aircraft (Capsada and Heinrich, 2018). This technique offers flexibility by enabling complex grid configurations with multiple moving components through the decoupling of the background grid from the grids of the moving components. In this way, the VWE allows to capture of the combination of all individual aerodynamic contributions (such as the combination of angle of attack, side-slip angle, angular rates, and control surface deflections), in contrast to the analytical models that can only make predictions of individual contributions derived from a limited set of data points. Additionally, it offers improved accuracy in simulating localized flow phenomena, such as flow separation, which lower-fidelity techniques like the vortex-lattice method (VLM) cannot adequately capture.

The VWE is then coupled with the AWE system dynamics and the model predictive control (MPC) capability from *AWEbox*, enabling geometry-resolved aero-servo simulations. This coupling approach is illustrated in Fig. 1. The rigid body motion and control surface deflection rates, computed by *AWEbox*, drive the motion of the aircraft component grids within the VWE. As these grids move, the VWE updates the flowfield and determines the resulting forces and moments, which are then fed back into *AWEbox*. This coupling is demonstrated by tracking the pre-optimized 1-loop power cycle in the VWE using an MPC-based controller. Additionally, the forces and moments derived from the VWE are compared to those predicted by the analytical aerodynamic model (AAM) embedded in *AWEbox*, highlighting the differences.

Figure 1. The aero-servo coupling approach.

The structure of this paper is outlined as follows. Section 2 introduces the reference aircraft studied. Section 3 describes the VWE. Section 4 covers the AWE system dynamics and control capabilities from *AWEbox*, explaining the trajectory optimization process and defining the controller used in this study. The coupling between the VWE and *AWEbox* is detailed through a stepwise approach in Sect. 5. Section 6 presents and discusses the results, and the conclusions are drawn in Sect. 7.

## 2 Reference aircraft

In this work, we consider a representation of an existing academic reference AWE aircraft, MegAWES (Eijkelhof and Schmehl, 2022). This aircraft has a wing area of  $150 \text{ m}^2$ , a wing span of 42.47 m, a root chord of 4.46 m, and a mass of 6885.2 kg. It is designed to have an electrical power output of up to 3 MW at  $22 \text{ m} \cdot \text{s}^{-1}$  wind speed. This aircraft consists of a wing, two ailerons, an elevator (all-moving horizontal tail), two rudders (all-moving vertical tail), and two fuselages. In this work, the focus is on the lifting surfaces, so the fuselages are omitted from the aerodynamic models.

Note that the geometry used here slightly differs from Eijkelhof and Schmehl (2022), specifically regarding the aileron location, which extends from 62.0% to 95.3% of the half-span. Additionally, the aileron gap is increased to 0.4 m to facilitate overset connectivity (as explained further below) with an allowable grid size for this simulation. There is also a rudder offset of 0.5 m from the elevator leading edge to prevent overlap between these components and to enable overset connectivity.

The center of gravity (CG) is located at [-1.67, 0, -0.229] m in the geometry axis system,  $[x, y, z]_G$ , located at the leading edge of the main wing as shown in Fig. 2.

**Figure 2.** (a) The MegAWES reference aircraft (Eijkelhof and Schmehl, 2022) and its representation in the VWE viewed from (b) the top and (c) the side. The values between the red brackets have been modified in this work, as the text explains.

## 3 Virtual wind environment

A VWE is constructed using a geometry-resolved CFD framework in *ANSYS Fluent*, incorporating 6-DOF rigid body motion and moving control surfaces. This makes the simulation suitable for analyzing the aerodynamics related to complex maneuvres and power cycles for airborne wind energy systems. This section first outlines the models and numerical settings employed. Subsequently, a detailed description of the aircraft grid, wind flow domain, and boundary conditions is provided. Finally, we explain the overset technique, which is used to connect the grids for various aircraft components and the background wind domain.

## 3.1 Flow model and numerical settings

The flow physics are modeled using the incompressible unsteady RANS equations with the k- $\omega$  SST model. Wall functions are used to model the boundary layer near the walls. Pressure–velocity coupling is achieved using a coupled scheme. Spatial and temporal discretization are implemented using a first-order upwind scheme for the convective terms in the momentum equations and a first-order implicit scheme with a timestep of 5 ms, respectively.

## 3.2 Aircraft component grids

For each lifting surface component, an individual structured grid with C-topology is constructed (Fig. 3). The grid domain of the main wing extends with a radius of 5 times the root chord in front of the wing, and the wake zone extends to 10 times the chord. The chord is divided into 132 cells, with refinements near the leading edge, trailing edge, and at the aileron location (at 75% chord) to enable overset connectivity with the aileron. This overset connectivity allows for interpolation of the flow variables between the separate grids as explained in Sect. 3.4. The wing domain is divided into 64 cells in the radial direction

with a growth rate of 1.15. The size of the first grid cells from the wall corresponds to a  $y^+$  value of approximately 50, which is within the valid range for wall functions between 30 and 500. While this approach reduces the accuracy in prediction flow separation, it provides a reasonable overall impression of the flow around the aircraft and limits the computational expense of the simulation. In the spanwise direction, the wing is discretized into 202 divisions, with 70 divisions at the location of each aileron. The grid of the wing is based on the grid refinement study that was performed in (Pynaert et al., 2023).

The C-grid radius and wake zone length of the elevator mesh are set to 1.75 and 3.5 times the elevator chord, respectively. For the rudder, these values are 1 and 2 times the rudder chord, respectively. The chords of both the rudder and elevator are divided into 152 cells, with 36 cells in the radial direction and a growth rate of approximately 1.2. For these components, the size of the first grid cells from the wall yields a  $y^+$  value of around 100. In the spanwise direction, both the elevator and rudder are discretized into 20 divisions. All aircraft component grids together consist of 6.3 million cells.

**Figure 3.** Cross section of the grids illustrating the C-topology for (a) the wing, (b) the ailerons, (c) the elevator, and (d) the rudders (Pynaert et al., 2024).

## 3.3 Wind flow domain and boundary conditions

A rectangular grid is constructed to simulate a wind flow domain of  $600 \times 600 \times 600$  m with a uniform cell size of 3 m (Fig. 4a). This rectangular grid comprises a total of 8 million cells. This work presents a proof of concept for a single power cycle simulation without accommodating wake development, making this domain size sufficient for the current purpose. However, a larger background size would be necessary for conducting wake studies, as demonstrated in (Haas et al., 2022; Crismer et al., 2024). The decomposition between the background grid and the aircraft grid, combined with the overset method (see Sect. 3.4), facilitates the simulation of the 6-DOF motion of the aircraft within a large computational domain while ensuring adequate refinement near the wall for assessing the aircraft's local aerodynamics.

The following boundary conditions are applied to the wind flow and aircraft domain boundaries to simulate an AWE system operating within the atmospheric boundary layer (ABL), as shown in Fig. 4. To model the ABL, a logarithmic velocity profile is used, expressed as:

$$v_{\mathbf{w}}(z) = \frac{u_*}{\kappa} \ln\left(\frac{z + z_0}{z_0}\right). \tag{1}$$

In this equation,  $u_* = 0.3829 \,\mathrm{m\cdot s^{-1}}$  is the friction velocity, representing the reference wind velocity scale,  $\kappa = 0.42$  is the Von Karman constant, z denotes the height, and  $z_0 = 0.0002 \,\mathrm{m}$  is the ground surface roughness height, characteristic of offshore conditions (Wieringa, 1992). The combination of  $u_*$ ,  $z_0$ , and  $\kappa$  corresponds to a reference wind speed of  $u_{\mathrm{ref}} = 12 \,\mathrm{m\cdot s^{-1}}$  at the reference height of  $z_{\mathrm{ref}} = 100 \,\mathrm{m}$ .

The specific dissipation rate ( $\omega$ ) profile is defined by Eq. (2), as proposed by Yang et al. (2009), to minimize the inconsistency between inlet profiles and the rough wall formulation in the k- $\omega$  SST model:

$$140 \quad \omega = \frac{u_*}{\kappa_* \sqrt{C_\mu}} \frac{1}{z + z_0}. \tag{2}$$

Here,  $C_{\mu}=0.09$  is a turbulence parameter. Although Yang et al. (2009) proposed a formulation for the turbulent kinetic energy (k) profile, it is not applied here due to limitations in defining it for the atmospheric boundary layer (ABL) conditions used. Instead, a default value of  $k=1~{\rm m^2 s^{-2}}$  is applied at the inlet. This value remains above  $k=0.7~{\rm m^2 s^{-2}}$  across the domain and reaches a maximum of  $k=1.75~{\rm m^2 s^{-2}}$  at ground level.

This logarithmic profile approximates the atmospheric surface layer and demonstrates the simulation's capability to incorporate specific wind profiles within the domain, which can be readily replaced with alternative profiles if needed. The entire domain is initialized using these inlet conditions. A uniform pressure of 1 atm is imposed at the outlet, and symmetry conditions are applied to the sides and top of the wind flow domain. The bottom boundary of the domain, representing the ground, is set as a stationary, no-slip wall with a roughness height  $z_0$  to align with the imposed logarithmic wind velocity at the inlet. A moving no-slip wall condition is applied to the surfaces of the aircraft (wing, ailerons, elevator, and rudders).

**Figure 4.** Complete flow domain with boundary conditions at (a) the wind flow domain and (b) the aircraft component domains (Pynaert et al., 2024).

## 3.4 Overset technique

155

160

The background grid (wind flow domain) and the various aircraft component grids are coupled using overset boundary conditions at the boundaries of the aircraft component domains. This method enables the simulation of the aircraft's rigid body motion, including deflected control surfaces, without deforming or re-generating the mesh. Figure 5 illustrates the connectivity between the background grid and the wing, elevator, and rudder grids in both the body-fixed xz-plane and xy-plane.

In the overset technique, specific cell types (donor and receptor) are assigned to cells at the overset boundary. The flow solution from donor cells is interpolated and transferred to the receptor cells of other components, while no interpolation is used for the other cells. To assign these cell types, the grid priority method is applied, giving the highest priority to the control surfaces - aileron, rudder, and elevator - followed by the main wing, with the background grid having the lowest priority. For components of equal priority, a boundary distance-based priority method is used. This ensures that the overset cells are positioned as far as possible from moving boundaries, which promotes solver convergence.

Figure 5. Overset cell types in the wing component grid in the (a) body-fixed xz-plane at y=1.3 m and in the (c) body-fixed xy-plane at z=0 m. Overset cell types in the aileron grid in the (b) body-fixed xz-plane at y=15 m. Overset cell types in the elevator component grid in the (d) body-fixed xy-plane at z=1 m. Overset cell types in the rudder components grid in the (e) body-fixed xy-plane at z=1 m. Calculated cells are highlighted in green, donor cells in red, and receptor cells in blue (Pynaert et al., 2024).

## 4 AWE system dynamics and control

To simulate realistic trajectories aimed at maximizing power output, the *AWEbox* toolbox (De Schutter et al., 2023) is employed. *AWEbox* provides capabilities for the modeling and optimal control of both single- and multi-aircraft AWE systems and is built on *CasADi*, a non-linear optimization framework using the algorithm differentiation tool *Autodiff* and inter-point optimizer *IPOPT*. This section outlines the specific capabilities from *AWEbox* that are utilized to build the aero-servo coupling. These

capabilities include the formulation of the AWE system dynamics and an AAM of the aircraft. Additionally, the periodic optimal control problem (POCP) formulation is employed to generate a reference trajectory for the megAWES aircraft. The final objective is to fly this trajectory within the VWE, utilizing the MPC toolbox for effective flight path tracking.

#### 170 4.1 AWE system dynamics

175

This work considers the AWE system using 6-DOF aircraft dynamics and assumes a straight tether with mass and drag. The dynamics are represented using two reference frames: a body-fixed reference frame,  $[x, y, z]_B$ , located at the CG of the aircraft, and an inertial frame,  $[x, y, z]_I$ , positioned at the ground station. In the body-fixed frame, the x-axis points to the rear of the aircraft, the z-axis points upward, and the y-axis extends towards the right wing, forming a right-handed coordinate system. In the inertial frame, the x-axis aligns with the wind direction, the z-axis points upward, and the y-axis completes the right-handed system. These coordinate systems are illustrated in Fig. 6. This paper uses the following convention: lowercase italic letters represent scalars, bold lowercase letters denote vectors, and bold uppercase letters indicate matrices.

The state variables  $\mathbf{x}$  of the system include the aircraft's position  $\mathbf{q}$  and the velocity  $\dot{\mathbf{q}}$  in the inertial frame, the direct cosine matrix (DCM)  $\mathbf{R}$ , representing the orientation of the aircraft, the angular velocity  $\boldsymbol{\omega}$  in the body-fixed frame, the aileron deflection  $\delta_{\mathbf{a}}$ , the rudder deflection  $\delta_{\mathbf{r}}$ , and the elevator deflection  $\delta_{\mathbf{e}}$ . The control surface deflections are grouped in  $\boldsymbol{\delta}$ . Additionally, the states include the tether length l, its reel-in/out speed l, and acceleration l. The control inputs  $\mathbf{u}$  to the system are the deflection rates of the control surfaces  $\dot{\delta}_{\mathbf{a}}$ ,  $\dot{\delta}_{\mathbf{r}}$ ,  $\dot{\delta}_{\mathbf{e}}$ , collected in  $\dot{\boldsymbol{\delta}}$ , as well as the tether reel-in/out jerk l.

The relevant system parameters  $\mathbf{p}$  include the wind speed  $v_{\rm w}(z)$ , defined by  $u_{\rm ref}$ ,  $z_{\rm ref}$ , and  $z_0$ , the aircraft mass  $m_{\rm W}$ , and the aircraft's inertia tensor  $\mathbf{J}$ , defined in the body-reference frame. Figure 6 provides a visualization of the key system states, controls, and parameters. A comprehensive overview of the AWE system parameters and constraints can be found in appendix  $\mathbf{B}$ .

**Figure 6.** (left) Visualization of the key states, controls, and parameters of the AWE system dynamics (Pynaert et al., 2024), and (right) the system state **x** and control input **u** vectors.

The system dynamics model employed in this study is based on the formulation presented in (Gros and Diehl, 2013), derived using Lagrangian mechanics. The resulting translation dynamics for a single aircraft is expressed as:

$$(m_{\rm W} + \frac{1}{3}m_{\rm T})\ddot{\mathbf{q}} + \lambda \mathbf{q} = \mathbf{f}_{\rm e,I} - (m_{\rm W} + \frac{1}{2}m_{\rm T})g\mathbf{l}_z.$$
 (5)

In this equation,  $\mathbf{l}_z = \begin{bmatrix} 0 & 0 & 1 \end{bmatrix}_{\mathrm{I}}^T$ , and  $\lambda$  is the algebraic Lagrange multiplier associated to the constraint c. The total external force  $\mathbf{f}_{\mathrm{e,I}}$ , expressed in the inertial frame I, acting on the system comprises the aerodynamic force of the aircraft,  $\mathbf{f}_{\mathrm{I}}$ , and the tether drag force  $\mathbf{f}_{\mathrm{T,I}}$ . The calculation of the tether drag is performed by dividing the tether into 5 segments, applying a multisegment drag model as described in (De Schutter et al., 2023). This model utilizes a constant tether drag coefficient,  $C_{\mathrm{D,T}}$ , set to 1.2. The tether mass,  $m_{\mathrm{T}}$ , is defined by:

195 
$$m_{\rm T} = \rho_{\rm T} l \frac{\pi D_{\rm T}^2}{4}$$
. (6)

In this equation,  $D_T$  and  $\rho_T$  are the tether diameter and density, respectively, whose values are given in appendix B.

The rotational dynamics is given by:

$$\mathbf{J}\dot{\boldsymbol{\omega}} = \mathbf{m}_{e,R} - \boldsymbol{\omega} \times \mathbf{J}\boldsymbol{\omega}. \tag{7}$$

The total external moment  $\mathbf{m}_{e,B}$ , expressed in the body-fixed frame B, is equal to the aircraft's aerodynamic moment,  $\mathbf{m}_{B}$ , without contribution from the tether, as the tether is attached to the CG. In the next section, we discuss the aircraft's aerodynamic forces  $\mathbf{f}_{I}$  and moments  $\mathbf{m}_{B}$  in depth.

The aircraft is constrained to ensure that the distance between the aircraft's CG and the origin matches the tether length, enforcing a straight tether:

$$c = \frac{1}{2}(\mathbf{q}^T\mathbf{q} - l^2) = 0.$$
 (8)

In Malz et al. (2019), it was found that the straight tether assumption is adequate for estimating power generation in a small-scale airborne wind energy (AWE) system (specifically, the AP2 developed by the former Ampyx Power). In contrast, the study of Heydarnia et al. (2024), based on the MegAWES aircraft, concluded that the straight-tether assumption can lead to the overestimation of harvested power up to 33%. Future work will focus on incorporating tether sag into the system dynamics model.

210 The DCM R contains the unit vectors of the body-fixed frame in the inertial frame. This non-minimal coordinate representation requires an orthonormality constraint:

$$c_R = \mathbf{P}_{ut}(\mathbf{R}^T \mathbf{R} - \mathbf{I}) = 0. \tag{9}$$

In this equation, I is the identity matrix and the operator  $P_{ut}$  is used to select the six upper triangular elements of a matrix (De Schutter et al., 2023).

Both the dynamics equations and the constraints c,  $c_R$  need to be enforced. An order reduction technique is applied to obtain an index-1 differential-algebraic equation by differentiating c twice with respect to time. Consistency conditions  $(c, \dot{c}, c_R) = 0$  must be enforced at an arbitrary time point in the trajectory. The system's kinematics are integrated in time using an explicit Euler scheme, which is consistent with the time integration of mesh movement in *ANSYS Fluent*. For a more detailed explanation of the AWE system dynamics and kinematics in *AWEbox*, the reader is referred to (De Schutter et al., 2023).

#### 220 4.2 Analytical aerodynamic model

To complete the dynamic model in AWEbox, the AAM proposed in (Malz et al., 2019) has been adapted for the MegAWES aircraft. This model is expressed by Eq. (11), where the superscript a refers to the AAM. Note that this model uses a different axis system, referred to as the aerodynamic axis system  $[x,y,z]_A$ , with the x-axis pointing forward and the z-axis pointing downward. In this equation,  $\rho(z)$  represents the air density and is modeled according to the international standard atmosphere (Archer, 2013), S is the wing surface area, and  $\mathbf{v}_a$  is the apparent wind velocity, which is a function of both the wind velocity  $v_w(z)$  (as described in Eq. (1)) and the aircraft velocity  $\dot{\mathbf{q}}$ :

$$\mathbf{v}_{\mathbf{a}} = [v_{\mathbf{w}}(z), 0, 0]^{T} - \dot{\mathbf{q}}. \tag{10}$$

The transformation matrix T transforms a vector in the aerodynamic frame to the body-fixed frame. The force coefficients  $C_x$ ,  $C_y$ , and  $C_z$  are functions of the angle of attack  $\alpha$ , sideslip angle  $\beta$ , roll rate p, pitch rate q, yaw rate r, and control surface deflections  $\delta_{a,e,r}$ , as described by Eq. (13). The roll moment coefficients  $C_l$ , the pitch moment coefficient  $C_m$ , and the yaw moment coefficient  $C_n$  are computed similarly. The stability derivatives  $C_{i,j}$  represent the contributions of the quantity

 $j = \{\alpha, \beta, p, q, r, \delta_{a,e,r}\}$  to the forces in the *i*-direction and the moments about the *i*-axis. These derivatives are computed with the aid of the VWE using simple flight maneuvers (see appendix A). The stability derivatives are then fitted to a second-order polynomial function of  $\alpha$ . The polynomial coefficients,  $c_2$ ,  $c_1$ , and  $c_0$ , are summarized in Tables A1 and A2.

Figure 7. Illustration of the aerodynamic properties relevant to the AAM.

235

$$\mathbf{f}_{\mathbf{I}}^{\mathbf{a}} = \frac{1}{2}\rho(z)\|\mathbf{v}_{\mathbf{a}}\|^{2}S\mathbf{R}\mathbf{T}\begin{bmatrix}C_{x}\\C_{y}\\C_{z}\end{bmatrix}, \mathbf{m}_{\mathbf{B}}^{\mathbf{a}} = \frac{1}{2}\rho(z)\|\mathbf{v}_{\mathbf{a}}\|^{2}S\mathbf{T}\begin{bmatrix}bC_{l}\\cC_{m}\\bC_{n}\end{bmatrix}$$
(11)

$$\mathbf{T} = \begin{bmatrix} -1 & 0 & 0 \\ 0 & 1 & 0 \\ 0 & 0 & -1 \end{bmatrix} \tag{12}$$

$$\begin{bmatrix} C_x \\ C_y \\ C_z \end{bmatrix} = \begin{bmatrix} C_{x,0} \\ C_{y,0} \\ C_{z,0} \end{bmatrix} + \begin{bmatrix} C_{x,\beta} \\ C_{y,\beta} \\ C_{z,\beta} \end{bmatrix} \beta + \begin{bmatrix} C_{x,p} & C_{x,q} & C_{x,r} \\ C_{y,p} & C_{y,q} & C_{y,r} \\ C_{z,p} & C_{x,q} & C_{x,r} \end{bmatrix} \begin{bmatrix} bp \\ cq \\ br \end{bmatrix} \frac{1}{2\|\mathbf{v}_a\|} + \begin{bmatrix} C_{x,\delta_a} \\ C_{y,\delta_a} \\ C_{z,\delta_a} \end{bmatrix} \delta_a + \begin{bmatrix} C_{x,\delta_c} \\ C_{y,\delta_c} \\ C_{z,\delta_c} \end{bmatrix} \delta_c + \begin{bmatrix} C_{x,\delta_r} \\ C_{y,\delta_r} \\ C_{z,\delta_r} \end{bmatrix} \delta_r \qquad (13)$$

While this model includes the primary aerodynamic effects required to simulate 6-DOF aircraft maneuvers, it relies on the following assumptions. The stability derivatives are calculated using CFD simulations at a constant flight speed of  $80 \text{ m} \cdot \text{s}^{-1}$ ,

and consequently, the Reynolds number is assumed constant during the computation of these derivatives. However, both the flight speed and the Reynolds number vary during flight. Additionally, the model assumes a linear relationship between  $\beta$ , p, q, r,  $\delta_{a,e,r}$ , and their effects on the force and moment coefficients. Furthermore, this model is quasi-steady and, therefore, independent of time, while in reality, unsteady aerodynamic effects occur. These limitations are addressed in the VWE, and the resulting differences are discussed in the results section.

## 4.3 Flight path generation

240

255

The reference flight path  $(\mathbf{x}^r(t), \mathbf{u}^r(t))$  considered in this work is generated by solving a POCP with a time period T, which is treated as an optimization variable. The POCP is formulated by Eqs. (15) - (18) (De Schutter et al., 2023). The objective function combines the power and penalties on the reference control actuation  $\mathbf{u}^r(t)$  to prevent actuator fatigue, on the sideslip angle  $\beta(t)$  to avoid side forces, and on angular accelerations  $\dot{\omega}(t)$  to prevent too aggressive maneuvers. These variables are collected in  $\hat{w}(t)$  and weighted by the matrix  $\mathbf{W}$ . The optimization variables include the reference system states  $\mathbf{x}^r(t)$ , the reference control inputs  $\mathbf{u}^r(t)$ , the reference algebraic Lagrange multiplier  $\lambda^r(t)$ , and the time period T. The power output of the system is determined by:

$$P(t) = F_{\mathsf{T}}(t)\dot{l}(t) = -\lambda(t)l(t)\dot{l}(t). \tag{14}$$

Here,  $F_T$  represents the tension of the tether. The AWE system dynamics (including the AAM) and kinematics, represented by  $\mathbf{F}$ , and the inequality constraints for path generation, represented by  $\mathbf{h}_g$ , must be satisfied at every time step. Equation 17 bundles the following constraints. First, constraints are applied to ensure that the flight envelope (angle of attack and sideslip angle) is not violated. Furthermore, constraints ensure that the maximum tether force is not exceeded. Finally, aircraft orientation constraints prevent collision between the aircraft and the tether. Additionally, bounds are imposed on flight altitude, tether length, speed, acceleration, aircraft angular velocity, control surface deflections and their rates, and the time period T. Finally, the reference initial state  $\mathbf{x}^{\mathbf{r}}(0)$  must be equal to the reference final state  $\mathbf{x}^{\mathbf{r}}(T)$  to enforce the periodicity of the trajectory (Eq. (18)).

$$\min_{\mathbf{x}^{\mathsf{r}}(t),\mathbf{u}^{\mathsf{r}}(t),\lambda^{\mathsf{r}}(t),T} \quad \frac{1}{T} \int_{0}^{T} \left( -P(t) + \hat{w}(t)^{T} \mathbf{W} \hat{w}(t) \right) dt \tag{15}$$

s.t. 
$$\mathbf{F}(\dot{\mathbf{x}}^{r}(t), \mathbf{x}^{r}(t), \mathbf{u}^{r}(t), \lambda^{r}(t), \mathbf{p}) = 0, \quad \forall t \in [0, T]$$
 (16)

$$\mathbf{h}_{\mathbf{g}}(\dot{\mathbf{x}}^{\mathbf{r}}(t), \mathbf{x}^{\mathbf{r}}(t), \mathbf{u}^{\mathbf{r}}(t), \lambda^{\mathbf{r}}(t), \mathbf{p}) \le 0, \quad \forall t \in [0, T]$$
(17)

$$\mathbf{x}^{\mathbf{r}}(0) - \mathbf{x}^{\mathbf{r}}(T) = 0 \tag{18}$$

## 4.4 Flight path tracking

The final capability used from the AWEbox toolbox is the MPC, which is used to track the reference flight trajectory. This controller, called at the current time  $\hat{t}_0$ , solves an optimal control problem during the simulation to steer the aircraft toward the reference flight path in an optimal manner. The optimal control formulation, with a moving time horizon  $T_h$ , is given by Eqs. (19) - (22) (Gros et al., 2013). The objective is to minimize the difference between the system states  $\mathbf{x}(t)$  and controls  $\mathbf{u}(t)$  over the upcoming time horizon, and the optimal reference states  $\mathbf{x}^r(t)$  and controls  $\mathbf{u}^r(t)$ , which are determined using the method described in the previous section. The weighting matrices  $Q_c$ ,  $R_c$ , and  $P_c$  are used to track the states, the controls, and the terminal cost, respectively. For this problem, equal weightings are assigned to each state and control variable. The optimization variables are the system states  $\mathbf{x}(t)$ , control inputs  $\mathbf{u}(t)$ , and the algebraic Lagrange multiplier  $\lambda(t)$ . Similar to the optimal control problem for flight path generation, the flight dynamics  $\mathbf{F}$  (including the AAM) and the path tracking constraints  $\mathbf{h}_t$  must hold over the MPC time horizon. The tracking constraints  $\mathbf{h}_t$  are more relaxed than the generation constraints  $\mathbf{h}_g$  to allow for more controllability, and they are summarized in appendix  $\mathbf{B}$ . The initial state  $\mathbf{x}(\hat{t}_0)$  is set equal to the current state estimate  $\hat{\mathbf{x}}_0$ .

$$\min_{\mathbf{x}(t), \mathbf{u}(t), \lambda(t)} \int_{\hat{t}_0}^{\hat{t}_0 + T_h} \left( ||\mathbf{x}(t) - \mathbf{x}^{\mathsf{r}}(t)||_{Q_c}^2 + ||\mathbf{u}(t) - \mathbf{u}^{\mathsf{r}}(t)||_{R_c}^2 \right) dt + ||\mathbf{x}(\hat{t}_0 + T_h) - \mathbf{x}^{\mathsf{r}}(\hat{t}_0 + T_h)||_{P_c}^2$$
(19)

s.t. 
$$\mathbf{F}(\dot{\mathbf{x}}(t), \mathbf{x}(t), \mathbf{u}(t), \lambda(t), \mathbf{p}) = 0, \quad \forall t \in [\hat{t}_0, \hat{t}_0 + T_h]$$
 (20)

$$\mathbf{h}_{t}(\dot{\mathbf{x}}(t), \mathbf{x}(t), \mathbf{u}(t), \lambda(t), \mathbf{p}) \leq 0, \quad \forall t \in [\hat{t}_{0}, \hat{t}_{0} + T_{h}]$$
(21)

$$\mathbf{x}(\hat{t}_0) = \hat{\mathbf{x}}_0 \tag{22}$$

To solve both the flight path generation and tracking optimal control problems, the problem formulations are transcribed to a nonlinear program (NLP) using direct collocation and solved using an interior-point homotopy (IPH) method. For a detailed description of the solution methods for the optimal control formulations, the reader is referred to (De Schutter et al., 2023).

The MPC's sample time is 5 ms, matching the timestep in the VWE, and the prediction horizon is  $T_{\rm h}=0.1$  s, corresponding to 20 sample times. In this case, the MPC is re-evaluated at each timestep. The current settings were established empirically to ensure simulation stability.

## 5 Aero-servo coupling

The previous sections introduced the VWE based on geometry-resolved CFD and the AWE system dynamics and control capabilities from *AWEbox*. This section details the coupling of these frameworks, referred to as the aero-servo coupling. An explicit coupling strategy is employed, meaning each solver is evaluated only once per timestep, with no coupling iterations performed within a single timestep. This approach introduces a small offset in the system's states for each solver and could theoretically lead to coupling instabilities, but this has not been observed in the simulations.

The coupling process consists of the following steps:

- 1. Transmit the rigid body motion of the aircraft, including the deflection of its control surfaces as calculated by *AWEbox*, to the VWE.
- 2. The aircraft's mesh is moved according to this rigid body motion and the control surface deflection rates in the VWE. The flow solver then computes the resulting airflow around the aircraft, which allows for determining the aerodynamic forces and moments acting on the aircraft.

3. These calculated forces and moments are sent back to AWEbox.

4. The aircraft's movement for the next timestep is updated, and subsequently, the controller determines the next control action. Figure 8 illustrates the different coupling steps for timestep *n*.

Steps 1 and 3 are managed by the coupling tool *CoCoNuT* (Delaissé et al., 2021). Step 2 takes place in the VWE and step 4 in *AWEbox*. Note that the tether is not part of the VWE, so there is no direct interaction between the tether states and this framework. The tether force is determined by the system dynamics, Eq. (5).

Figure 8. Overview of the explicit aero-servo coupling for timestep n to go to timestep n+1 (for  $n \in [0, n_{\text{end}}-1]$ ). The numbers in brackets indicate the steps as explained in the text.

At initialization (step 0), the states and controls at t = 0 s from the *AWEbox* output files, obtained by the flight path generation, are used to initialize the solvers. To set up the VWE, the control surface component grids are deflected with  $\delta_0$ , and the aircraft is rotated using  $\mathbf{R}_0$  to establish the initial attitude. The aircraft is then positioned at  $\mathbf{q}_0$  and the flow is initialized with the wind field. At this stage, the aircraft is stationary, so the flow must first develop before it becomes meaningful. The dynamics and control in *AWEbox* are initialized with the states and controls at time t = 0 s, including the motion. The controller is then activated to determine the control action for the first time step.

## 305 5.1 Transmit rigid body motion (step 1)

The mesh motion of the aircraft components - the wing (w), the ailerons (a), the rudders (r), and the elevator (e) - is achieved using the zone motion function in *ANSYS Fluent*. This function requires the motion to be defined in terms of a translational velocity  $\mathbf{v}_i$ , with  $i \in \{w, e, r, a\}$ , a rotational velocity  $\omega_i$ , a rotation-axis origin  $\mathbf{o}_i$ , and a rotation-axis direction  $\mathbf{a}_i$  expressed in the inertial frame (see Fig. 9, left). The rotation-axis origin  $\mathbf{o}_i$  for each component i is the position of the aircraft's CG  $\mathbf{q}_n$ .

Figure 9. (left) Visualisation of the mesh motion parameters of the wing and (right) the aircraft's control surface hinges.

Defining the movement of the control surfaces requires special attention, as their deflection motion must be combined with the aircraft's overall motion. The deflection introduces an additional velocity component,  $\mathbf{r}_i \times \dot{\delta}_i \mathbf{h}_i$ , when expressed relative to the aircraft's origin, and the total velocity is given by Eq. (23). The wing deflection rate  $\dot{\delta}_w$  equals 0, so the second term drops for the wing. For the control surfaces, the vectors  $\mathbf{r}_i$  and  $\mathbf{h}_i$  represent the hinge positions and axis of the left aileron, right aileron, elevator, left rudder, and right rudder, respectively, relative to  $\mathbf{o}_i$  as shown in Fig. 9 (right). The angular motion due to the deflection rate is also added to the aircraft's angular motion and given by Eq. (24).

$$\mathbf{v}_i = \dot{\mathbf{q}}_n + \mathbf{R}_n(\mathbf{r}_i \times \dot{\delta}_{i,n} \mathbf{h}_i), \text{ for } i = \{w, e, r, a\}.$$
(23)

$$\boldsymbol{\omega}_{i,I} = \mathbf{R}_n(\boldsymbol{\omega}_n + \dot{\delta}_{i,n}\mathbf{h}_i), \text{ for } i = \{w, e, r, a\}.$$

The rotational velocity  $\omega_i$  and the rotation-axis direction  $\mathbf{a}_i$  can then be calculated as follows:

320 
$$\omega_i = \|\boldsymbol{\omega}_{i,I}\|, \text{ for } i = \{w, e, r, a\},$$
 (25)

$$\mathbf{a}_i = \frac{\omega_{i,1}}{\omega_i}, \text{ for } i = \{w, e, r, a\}.$$
 (26)

Note that each component is moved independently, and they are not linked to one another. Therefore, using the same time integration scheme in both *AWEbox* and *ANSYS Fluent* is necessary to prevent drift between the components.

## 325 5.2 Solving the flow with grid movement (step 2)

The grid of the components is moved according to the zone motion function defined in the previous step. This grid movement is taken into account in the convective fluxes of the flow's transport equations. For example, in the continuity equation, this can be expressed as:

$$\frac{d}{dt} \int_{V} \rho \, dV + \int_{S} \rho(\mathbf{v} - \mathbf{v}_{b}) \cdot \mathbf{n} dS = 0. \tag{27}$$

In this equation,  $\mathbf{v}$  is the flow velocity,  $\mathbf{v}_b$  is the grid velocity, V and S the volume and surface of each control volume respectively, and  $\mathbf{n}$  the normal vector to the surface. After solving the flow equations, the pressure distribution is integrated, and the resulting forces  $\mathbf{f}_{1,n+1}^f$  and moments  $\mathbf{m}_{1,n+1}^f$  (around the origin of the inertial frame) are exported.

#### 5.3 Feed back the forces and moments (step 3)

340

The VWE provides the forces and moments in the inertial frame I. While the toolbox AWEbox requires the forces in the inertial frame, it expects the moments in the body-fixed frame B, so the moments are transferred using Eq. (28). For this procedure, the location around which the moment is taken must correspond to the CG. Since the aircraft is first moved in the VWE, its position  $\mathbf{q}_{n+1}$  and attitude  $\mathbf{R}_{n+1}$  for the next time step are computed using the explicit Euler method (Eqs. (29) and (30)). These updated values are then used to determine the moments.

$$\mathbf{m}_{\mathrm{B},n+1}^{\mathrm{f}} = \mathbf{R}_{n+1}^{T} \left( \mathbf{m}_{I,n+1}^{\mathrm{f}} - \mathbf{q}_{n+1} \times \mathbf{f}_{I,n+1}^{\mathrm{f}} \right)$$
(28)

 $\mathbf{q}_{n+1} = \mathbf{q}_n + \dot{\mathbf{q}}_n \Delta t \tag{29}$ 

$$\mathbf{R}_{n+1} = \mathbf{R}_n + \dot{\mathbf{R}}_n \Delta t = \mathbf{R}_n + \mathbf{R}_n \boldsymbol{\omega}_n \times \Delta t \tag{30}$$

#### 5.4 Solving the system dynamics and control (step 4)

During the startup of the simulation (when  $n < n_1 = 220$ ), the forces  $\mathbf{f}_{\mathrm{I},n}^{\mathrm{a}}$  and moments  $\mathbf{m}_{\mathrm{B},n}^{\mathrm{a}}$  from the AAM (Eqs. (11) and (13)) are used, as the flow is still building up in the CFD solver. When  $n_1 = 220 < n < n_2 = 440$ , there is a transition period during which a weighted average is taken between the forces  $\mathbf{f}_{\mathrm{I},n+1}^{\mathrm{f}}$  and moments  $\mathbf{m}_{\mathrm{B},n+1}^{\mathrm{f}}$  from the VWE and the AAM:

$$\mathbf{f}_{\mathbf{I}} = (w-1)\mathbf{f}_{\mathbf{I},n}^{\mathbf{a}} + w\mathbf{f}_{\mathbf{I},n+1}^{\mathbf{f}} \text{ and}$$
(31)

350 
$$\mathbf{m}_{\mathbf{B}} = (w-1)\mathbf{m}_{\mathbf{B},n}^{\mathbf{a}} + w\mathbf{m}_{\mathbf{B},n+1}^{\mathbf{f}}, \text{ for } n_1 < n < n_2.$$
 (32)

The weight w of the VWE forces and moments varies linearly from 0 at  $n=n_1$  to 1 at  $n=n_2$ . After  $n>n_2$ , only the forces and moments from the VWE are considered. This timestep corresponds to  $2.2 \, \mathrm{s}$ , after which the starting vortex is sufficiently distant and startup effects are considered negligible. Using these forces and moments, the system of differential-algebraic equations (Sect. 4.1) is solved for  $\ddot{\mathbf{q}}_n$  and  $\dot{\boldsymbol{\omega}}_n$  and these values are filled in the state derivative vector  $\dot{\mathbf{x}}_n = \left\{\dot{\mathbf{q}}, \ddot{\mathbf{q}}, \dot{\mathbf{R}}, \dot{\boldsymbol{\omega}}, \dot{\boldsymbol{\delta}}, \dot{l}, \ddot{l}, \ddot{l}\right\}_n$  together with the control inputs  $\dot{\delta}_n$  and  $\ddot{l}_n$ , previously determined from the MPC.

The explicit Euler scheme is then used to update the states for the next timestep. This can be expressed as:

$$\mathbf{x}_{n+1} = \mathbf{x}_n + \dot{\mathbf{x}}_n \Delta t. \tag{33}$$

Finally, the MPC is used to calculate the control inputs  $\dot{\delta}_{n+1}$  and  $\ddot{l}_{n+1}$  for the new states, using the method explained in Sect. 4.4.

360 This loop (steps 1-4) continues for each new timestep, starting again with step 1.

#### 6 Results

355

365

We simulate a 1-loop crosswind flight with the MegAWES aircraft in a wind field representative of offshore conditions as a demonstration for the aero-servo coupling. First, we present the optimized reference trajectory for this wind condition and aircraft parameters. Then, we present results from a simulation in which the VWE and AWE system dynamics are fully coupled, tracking the reference flight path within the VWE. Finally, we present a qualitative analysis of the flow field for this simulation and summarize the computational time required for the simulation.

#### 6.1 Optimized reference flight path

The optimized reference trajectory, generated by the POCP detailed in Sect. 4.3 and based on the AAM, is illustrated in Fig. 10. The corresponding optimization parameters are summarized in Table B1. The aircraft starts at the top of the flight path, entering the reel-out phase (red). During this phase, the aircraft flies at an angle of attack of 4°, and the aircraft descends, causing both speed and power output to increase. The power reaches a plateau at the maximum value of 2.5 MW (put as a constraint), where it remains for approximately 5 s until the aircraft reaches the bottom of the flight path. As the aircraft ascends, the angle of attack, speed, and power output decrease, and it transitions into the reel-in phase (blue), where power is consumed. At its peak, the power required to reel-in the aircraft is approximately 1.8 MW. This 1-loop power cycle takes 20.0 s to complete and produces an average power output of 436 kW.

**Figure 10.** (left) Visualisation of the optimized reference trajectory. The grey aircraft is shown to scale at its initial position, connected by the black tether to the ground station, which is located at the origin of the inertial frame. The inlet wind field is depicted in blue, and the dashed box indicates the simulation domain of the VWE. (right) Power and airspeed plotted over time.

## 6.2 Tracking the reference trajectory

In this section, we demonstrate the aero-servo coupling, as outlined in Sect. 5, by tracking the reference trajectory in the VWE using MPC. From Fig. 11, we observe that the reference trajectory is tracked with high accuracy in terms of position, with a maximum deviation of 4.0 m occurring at the bottom of the loop and a root mean square deviation of 1.6 m over the cycle. The power curve is tracked with moderate accuracy, with the largest deviation of 442 kW occurring during the transition from reel-out to reel-in. The root mean square deviation over the cycle amounts 93 kW and the reduction in average power is 16 kW, which represents 4% of the reference average power.

**Figure 11.** (left) The reference trajectory (REF) and the trajectory in the coupled simulation (COSIM). (right) The reference power (REF) and the resulting power from the coupled simulation (COSIM) over time.

The aerodynamic properties of this coupled simulation (COSIM), more specifically the angle of attack  $\alpha$  (a), the side-slip  $\beta$  (b), and the apparent wind speed  $V_a$  (c), are plotted in the left column of Fig. 12. The apparent wind speed remains within  $0.5~{\rm m\cdot s^{-1}}$  from the reference values, while the angle of attack, and especially the side-slip angle deviate up to  $2.5^{\circ}$  and  $5^{\circ}$ , respectively, from the reference trajectory. This increased deviation arises because the side-slip angle  $\beta$  is not directly associated with any aircraft state explicitly tracked by the MPC. The resulting aerodynamic forces from the VWE (blue), shown in the right column of Fig. 12 in the body-fixed frame, exhibit oscillations around the reference forces, with a maximum deviation of  $0.16~{\rm for}~C_z$ ,  $0.02~{\rm for}~C_y$ , and  $0.04~{\rm for}~C_x$ , at  $t=9.0~{\rm s}$ . Future work could focus on control strategies to minimize these oscillations, thereby reducing structural stresses and extending remaining useful life (RUL).

A comparison is now made between the resulting forces obtained from the VWE and those predicted by the AAM, the model employed within the MPC framework. The resulting forces from the AAM and VWE exhibit a consistent trend. The maximum force deviations are 0.080 for  $C_z$ , 0.007 for  $C_y$ , and 0.008 for  $C_x$ , and the root mean square deviation over the cycle is 0.025 for  $C_z$ , 0.002 for  $C_y$ , and 0.004 for  $C_x$ . The CFD framework used to derive the stability derivatives for the AAM also forms the foundation of the VWE. Therefore, the remaining discrepancies likely arise from the AAM's limitations in capturing nonlinear aerodynamic effects beyond the angle of attack, as well as unsteady aerodynamic phenomena.

**Figure 12.** (left) The aerodynamic properties from the reference trajectory (REF) and the coupled simulation (COSIM, properties hold for both VWE and AAM). (right) The aerodynamic forces, expressed in the body-fixed frame, resulting from the VWE and predicted by the AAM.

In the left column of Fig. 13, the control surface deflection of (a) the aileron  $\delta_a$ , (b) the elevator  $\delta_c$ , and (c) the rudders  $\delta_r$  are plotted. In addition, the angular rates (d) p, (e) q, (f) r are shown. The control surface deflections and angular rates in the simulation generally follow the reference trajectory. The largest deviation is observed in the rudder deflection, reaching approximately  $8^{\circ}$  at t=5.0 s. The rudder deflection also hits the constraint of  $-10^{\circ}$  during the simulation. Note that this constraint for trajectory generation is  $\pm 5^{\circ}$ . Meanwhile, the maximum deviation for the angular rates amounts to approximately  $9^{\circ}s^{-1}$  for the pitch rate at t=8.6 s. The control inputs are more aggressive than the reference, leading to greater oscillations in

the aerodynamic moments around the reference value, with a maximum deviation of 0.08 for the pitch moment  $C_m$  coefficient at t = 8.6 s, as seen in the right column of Fig. 13.

**Figure 13.** (left) Control surface deflections and (middle) rotation rates from the reference trajectory (REF) and the coupled simulation (COSIM). (right) The aerodynamic moments, expressed in the body-fixed frame, from the VWE and predicted by the AAM.

The moments predicted by the AAM and VWE generally follow the same global trend. The maximum offset between the two models amounts to 0.01 for the roll  $C_l$ , 0.03 for the pitch  $C_m$ , and 0.004 for the yaw moment coefficient  $C_n$ , respectively. The root mean square deviation between the two models over the cycle amounts to 0.006 for the roll  $C_l$ , 0.01 for the pitch  $C_m$ , and 0.001 for the yaw  $C_n$  moment coefficient, respectively. Because the roll moment coefficient  $C_l$  is normalized with the span b, in contrast with the pitch  $C_m$  that is normalized with the chord c, the offset in the roll moment is the largest in magnitude.

This discrepancy in the moment coefficients arises due to the complex combination of multiple aerodynamic contributions, influenced by the highly dynamic motion of the aircraft. For example, the yaw rate r of the aircraft induces an asymmetric lift distribution, generating a roll moment. The ailerons are deflected to counteract this moment. In the AAM, these aerodynamic contributions are combined linearly. In contrast, the yaw rate r in the VWE also impacts aileron effectiveness, a phenomenon not captured in the AAM. Specifically, the rotational motion increases aileron effectiveness because the outer aileron, which experiences less flow separation (as further discussed in the next section), encounters a higher apparent wind speed. This interaction accounts for the observed offset in the AAM's predictions. Furthermore, also the pitch and yaw moments are significantly influenced by rotational rates, further underscoring the limitations of the AAM. These observations highlight the necessity of using a full CFD-based approach, as implemented in the VWE, to accurately capture the complex aerodynamic interactions.

## 420 6.3 Qualitative analysis of the flow field

The VWE provides the velocity and pressure of the flow field throughout the trajectory. This data is valuable for analyzing the design and operation of future AWE systems. In this section, we demonstrate some key flow features. Figure 14 shows the pressure coefficient distribution on the aircraft at two different times: t = 5 s and t = 15 s, which correspond to the highest and lowest power points in the cycle. By using the pressure distribution on the wing, the local lift distribution (represented by  $C_z$ ) can be derived, as shown in Fig. 15 at four different time instances. The drop in lift is attributed to the aileron gap. Furthermore, it is evident that the lift distribution is asymmetric, a result of the rotational motion. The ailerons are deflected to compensate for this asymmetry. However, while the left aileron is deflected downward, it does not increase the local lift coefficient due to flow separation on the aileron, as seen in Fig. 16 (c). The resulting lift distribution deviates from the ideal elliptical lift distribution, which maximizes performance for conventional aircraft. While the ideal lift distribution for AWE systems is not yet well-established, it is clear that improvements are needed for this aircraft design to optimize performance.

**Figure 14.** Pressure distribution at t = 5 s and t = 15 s.

**Figure 15.** Spanwise  $c_z$  distribution at four different time instances.

A contour plot of the apparent wind velocity at  $t=5~\rm s$  is shown in Fig. 16, providing a visualization of the flow field around all lifting surfaces. As previously mentioned, the left aileron experiences flow separation, with the right aileron also showing slight separation. Additionally, the rear portion of the main wing, located just ahead of the aileron, exhibits separated flow. This highlights the need for an improved aileron design to enhance the performance and controllability of the system and demonstrates the ability of the VWE to assess these flow phenomena for the whole power cycle. The main wing also displays slight flow separation, impacting the elevator, which operates in its wake. The flow around the elevator and rudders remains attached. Despite the equal deflection of the rudders, the flow field around them is not symmetric due to interactions between the rudders and the circular motion of the aircraft.

Figure 16. Contour plots of the apparent velocity magnitude at t=5 s. The plots show the body-fixed xz-plane at (a) y=0 m (covering the wing and elevator), at (b) y=15 m (right aileron), at (c) y=-15 m (left aileron), and the (d) xy-plane at z=2 m (rudders).

The wall shear stress in the  $x_B$ -direction along the aircraft surfaces is examined, where negative values indicate flow reversal and signal regions of flow separation. Figure 17 presents this parameter at t = 5 s and t = 15 s. It is observed that the trailing

edge of the main wing and the entire left aileron exhibit consistent flow separation throughout the trajectory. These findings provide valuable insights for enhancing the aircraft's design to optimize performance across the complete power cycle.

Figure 17. Visualization of negative wall shear stress in the body-fixed x-direction (red) as an indication of separated flow regions at t = 5 s and t = 15 s.

## 6.4 Notes on computational time

The simulation was performed on a system with 2 × 20-core Intel Xeon Gold 6242R processors (3.1 GHz) and 187.4 GB of system memory. The peak memory usage during the simulation reached 92.2 GB. A breakdown of the computational time for one power loop cycle is provided in Table 1. Furthermore, it is calculated that 63% of the Fluent solver calculation time is attributed to the overset technique. It can also be observed that a substantial portion of the total simulation time was spent on data saving, indicating a potential area for optimization in future runs.

The MPC algorithm required approximately 0.6 s of wall time per time step. Although this is relatively efficient in the context of the full simulation, it exceeds the simulation time step of 5 ms, making this control configuration unsuitable for real-time implementation. However, the calculation time of the MPC depends on the implementation and hardware, and faster implementations are available, such as *acados* (Verschueren et al., 2020).

**Table 1.** Computational time of a 1-loop power cycle.

| Component                        | # of Cores | Hours | Per time step (s) |
|----------------------------------|------------|-------|-------------------|
| Total run time (4000 time steps) |            | 174.6 | 157.1             |
| Fluent solver                    | 40         | 130.4 | 117.4             |
| Coupling                         | 1          | 2.9   | 2.6               |
| Save time (coupling)             | 1          | 41.3  | 37.2              |
| MPC wall time                    | 1          | 0.7   | 0.6               |

## 7 Conclusion and outlook

455

460

This study introduces a comprehensive approach that couples a virtual wind environment (VWE), represented by geometry-resolved computational fluid dynamics (CFD), with the airborne wind energy (AWE) system dynamics and control toolbox, *AWEbox* (De Schutter et al., 2023), to enable aero-servo simulations for AWE systems. The aero-servo coupling is demonstrated by tracking a pre-optimized 1-loop reference trajectory for the MegAWES aircraft (Eijkelhof and Schmehl, 2022) in the VWE using the model predictive controller (MPC) from *AWEbox*. The simulation achieved 96% of the reference power with a maximum trajectory deviation of 4 m. We compared the resulting forces and moments against predictions from an analytical aerodynamic model (AAM). The comparison revealed consistent trends, although deviations were observed due to aerodynamic effects not captured by the quasi-steady AAM, such as the nonlinear contribution of the rotational motion on the control surfaces' effectiveness and force/moment coefficients. These findings underscore the importance of employing full CFD simulations.

This analysis has highlighted key flow characteristics, such as flow separation during crosswind flight maneuvers, to inform potential design and operational improvements. Enhancing the aileron design could help prevent flow separation, thereby increasing the control authority of these surfaces and boosting the overall system performance. The simulation revealed a significant interaction between the rotational motion of the aircraft, which is common in crosswind flight, and aileron effectiveness. These insights can be further studied and used to refine the AAM, ultimately reducing the model mismatch in the MPC and improving the controller's effectiveness.

Despite its capabilities, the current framework also exhibits certain limitations. First, it should be noted that full-geometry CFD simulations are significantly more computationally demanding, rendering them impractical for rapid design iterations. To limit the computational expense of the simulations presented in this work, the boundary layer at the aircraft surface is not fully resolved; instead, wall functions are employed. While this approach reduces the accuracy in predicting flow separation, it provides a reasonable overall impression of the flow around the aircraft. Nevertheless, the current framework is compatible with further mesh refinement. Furthermore, the current CFD mesh arrangement supports only first-order discretization, which may further limit simulation accuracy. The use of higher-order schemes is recommended to enhance the precision of the CFD results. A validation study is essential to establish the credibility of the simulation outcomes. Secondly, the wing deformation and fluid–structure interaction (FSI) effects are neglected in this study, based on the findings of Pynaert et al. (2023), which showed that structural deformations of the investigated aircraft were minimal. Specifically, the FSI analysis indicated only a 1.4% reduction in aerodynamic loads. Nevertheless, the framework has been designed to support the inclusion of FSI in future studies involving more flexible aircraft configurations. Finally, it is worth noting that the MPC controller is currently executed at every timestep; it is recommended to explore lower-frequency MPC evaluations.

This work represents a step forward in the ongoing development of a comprehensive aero-servo-elastic coupling, building upon prior advancements in aeroelastic modeling (Pynaert et al., 2023), which will be integrated into the current approach.

Notably, although this work focuses on ground-gen systems, the framework can be adapted for fly-gen systems with necessary modifications.

Author contributions. Methodology, N.P., T.H., J.W., G.C., J.D.; Formal analysis, N.P.; Project administration, J.D.; Resources, J.D.; Supervision, T.H., J.W., G.C., J.D.; Writing - original draft, N.P.; Writing - review and editing, T.H., J.W., G.C., J.D.; Funding acquisition, N.P., J.W., G.C., J.D. All authors have read and agreed to the published version of the manuscript.

490 *Competing interests.* The authors declare that they have no known competing financial interests or personal relationships that could have appeared to influence the work reported in this paper.

Acknowledgements. This work was conducted as part of the BORNE project (Belgian Offshore aiRborne wind Energy), funded by the Energy Transition Funds (ETF) from the FPS Economy. The authors acknowledge Dylan Eijkelhof and Roland Schmehl for providing the necessary data of the reference model, more specifically to replicate the aeroshell of the reference model. We extend our gratitude to Jochem De Schutter for his valuable assistance with the *AWEbox* toolbox. The computational resources in this work were provided by the VSC (Flemish Supercomputer Center). During the preparation of this work, the authors used *ChatGPT* to improve the flow and writing quality of the scientific text. After using this tool, the authors reviewed and edited the content as needed and took full responsibility for the content of the publication.

#### A Stability derivatives

495

This section outlines the calculation of the MegAWES stability derivatives using CFD. Three distinct setups, illustrated in Fig. A1, are employed for these calculations. Each setup uses identical numerical settings and the same aircraft grid as detailed in Sect. 3. All simulations for stability derivative calculations are conducted with an airspeed  $V_a$  of 80 m·s<sup>-1</sup>. The first and second setups involve steady-state calculations, and the third transient.

Figure A1. Simulation setups to calculate the stability derivatives.

The first setup focuses on calculating the stability derivatives associated with the control surface deflections  $\delta_{a,e,r}$  and side-slip angle  $\beta$ . In this simulation, the front and sides of the main wing grid serve as the inlet, while the back functions as the outlet. The airspeed of the aircraft, at the specified angle of attack and side-slip angle, is applied at the inlet.

The second setup is used to calculate the stability derivative related to the roll rate p. In this simulation, a small background grid is used, with overset connectivity to the aircraft grid. The roll motion is introduced by applying the corresponding frame motion to all components of the aircraft. The aircraft's airspeed, at the specified angle of attack, is applied at the inlet.

The third setup is used to calculate the stability derivative related to the main contribution of the angle of attack  $\alpha$ , the yaw rate r, and pitch rate q. This setup is similar to the VWE described in Sect. 3, but with a smaller background and zero wind velocity at the inlet. The aircraft moves according to the method described in step 1 from Sect. 5. Three flight maneuvres are considered: the first is a descending, ascending and horizontal straight flight maneuver to simulate a positive, negative, and zero angle of attack, respectively. The aircraft's straight motion with angle of attack  $\alpha$  is described by:

$$\omega = \begin{bmatrix} 0 \\ 0 \\ 0 \end{bmatrix}, \dot{\mathbf{q}} = \begin{bmatrix} -V_{\mathrm{a}} \cos \alpha \\ 0 \\ -V_{\mathrm{a}} \sin \alpha \end{bmatrix}.$$
 (A1)

The second and third flight maneuvers are a pure yawing flight with yaw rate r and a pure pitching flight with pitch rate q. These maneuvres are illustrated in Fig. A2. The aircraft's motion is described by the following equations for the yaw motion:

$$\boldsymbol{\omega}_{r} = \begin{bmatrix} 0 \\ 0 \\ r \end{bmatrix}, \dot{\mathbf{q}}_{r} = \begin{bmatrix} -V_{a}\cos\alpha\cos\psi \\ -V_{a}\cos\alpha\sin\psi \\ -V_{a}\sin\alpha \end{bmatrix}, \tag{A2}$$

and the pitch motion (assuming small angles  $\alpha$  and  $\theta$ ):

$$\omega_q = \begin{bmatrix} 0 \\ q \\ 0 \end{bmatrix}, \dot{\mathbf{q}}_q = \begin{bmatrix} -V_a \cos \alpha \cos \theta \\ 0 \\ -V_a (\sin \alpha - \sin \theta) \end{bmatrix}.$$
 (A3)

**Figure A2.** (a) The pure r-motion and (b) q-motion (Mulder et al., 2013).

In total, 53 simulations are performed to calculate the stability derivatives. For the main angle of attack contribution  $C_{i,0}$ , the angle of attack is varied between  $-10^{\circ}$  and  $10^{\circ}$  in steps of  $5^{\circ}$ . For the other contributions, the angle of attack is varied between  $-5^{\circ}$  and  $5^{\circ}$  in steps of  $5^{\circ}$ . For each angle attack the side-slip  $\beta$  and control surface deflections  $\delta_{\rm a,e,r}$  are set to a value of  $5^{\circ}$  and  $10^{\circ}$ . Additionally, the elevator deflection  $\delta_{\rm e}$  is set to  $-5^{\circ}$  and  $-10^{\circ}$ . A value of  $10^{\circ}s^{-1}$  and  $20^{\circ}s^{-1}$  is used for the contribution of rotation rates.

Because each simulation considers the variation of only 1 contribution j, while all other contributions are set to zero, Eq. (13) can be reformulated as follows to calculate the stability derivatives:

$$C_{i,j} = \frac{C_i - C_{i,0}}{i},\tag{A4}$$

for  $j = \beta$  and  $\delta_{a,e,r}$ ,

$$C_{i,j} = \frac{2V_{a}(C_i - C_{i,0})}{bj},$$
 (A5)

for j = p and r, and

$$C_{i,j} = \frac{2V_{\rm a}(C_i - C_{i,0})}{cj},$$
 (A6)

for j = q.

Here, the stability derivatives  $C_{i,j}$  collect the contributions of the quantity j to the forces in the i-direction, and the moments along the i-axis. The stability derivatives are subsequently fitted to a second-order polynomial function of  $\alpha$ :

$$C_{i,j} = \begin{bmatrix} c_2 & c_1 & c_0 \end{bmatrix} \begin{bmatrix} \alpha^2 \\ \alpha \\ 1 \end{bmatrix}. \tag{A7}$$

The resulting coefficients are given in Table A1 for the forces and Table A2 for the moments. *AWEbox* does not use the greyed-out values.

| Coefficient                 | $c_0$   | $c_1$   | $c_2$    |
|-----------------------------|---------|---------|----------|
| $C_{x,0}$                   | -0.1164 | 0.4564  | 2.3044   |
| $C_{x,b}$                   | 0.0279  | 0.0414  | 0.8307   |
| $C_{x,p}$                   | 0.0342  | 0.1529  | -1.8588  |
| $C_{x,q}$                   | -0.4645 | 8.5417  | -10.8181 |
| $C_{x,r}$                   | -0.0006 | 0.0519  | 0.4025   |
| $C_{x,\delta_{\mathtt{a}}}$ | -0.0168 | 0.0733  | 1.3335   |
| $C_{x,\delta_{e}}$          | 0.0002  | -0.0182 | 0.4100   |
| $C_{x,\delta_{	ext{r}}}$    | -0.0173 | -0.0150 | -0.2922  |
| $C_{y,0}$                   | -0.0000 | 0.0002  | 0.0013   |
| $C_{y,b}$                   | -0.2740 | 0.1664  | 0.8803   |
| $C_{y,p}$                   | 0.0198  | -0.2312 | -0.3150  |
| $C_{y,q}$                   | 0.0007  | -0.0010 | 0.0799   |
| $C_{y,r}$                   | 0.0911  | -0.0267 | -0.4982  |
| $C_{y,\delta_{\mathrm{a}}}$ | 0.0063  | 0.0119  | -0.0754  |
| $C_{y,\delta_{e}}$          | 0.0001  | -0.0012 | -0.0216  |
| $C_{y,\delta_{ m r}}$       | 0.2259  | -0.1198 | 0.1955   |
| $C_{z,0}$                   | -0.9245 | -3.7205 | 4.7972   |
| $C_{z,b}$                   | 0.1123  | -0.1250 | -5.0971  |
| $C_{z,p}$                   | 0.1387  | 0.1685  | -27.9934 |
| $C_{z,q}$                   | -5.6405 | 60.9970 | 240.6406 |
| $C_{z,r}$                   | 0.0067  | 0.1349  | -4.4412  |
| $C_{z,\delta_{\mathrm{a}}}$ | 0.0638  | -1.8662 | -26.6776 |
| $C_{z,\delta_{ m e}}$       | -0.4897 | 0.2366  | 3.4195   |
| $C_{z,\delta_{ m r}}$       | 0.0044  | 0.0123  | -0.2717  |

 Table A1. Aerodynamic force stability derivatives.

| Coefficient                 | $c_0$   | $c_1$   | $c_2$    |
|-----------------------------|---------|---------|----------|
| $C_{l,0}$                   | 0.0000  | 0.0002  | 0.0001   |
| $C_{l,b}$                   | 0.0344  | -0.1786 | -2.6711  |
| $C_{l,p}$                   | -0.4052 | 0.4109  | -0.5721  |
| $C_{l,q}$                   | 0.0180  | 0.0258  | -2.1828  |
| $C_{l,r}$                   | 0.1802  | 0.5792  | -0.0129  |
| $C_{l,\delta_a}$            | -0.0941 | -0.1921 | -0.2034  |
| $C_{l,\delta_{e}}$          | 0.0000  | -0.0063 | -0.0912  |
| $C_{l,\delta_{	ext{r}}}$    | 0.0106  | -0.0214 | -0.0874  |
| $C_{m,0}$                   | 0.0279  | -0.5307 | -0.9786  |
| $C_{m,b}$                   | -0.0184 | 0.7392  | 8.2241   |
| $C_{m,p}$                   | 0.0008  | -0.1007 | -0.0845  |
| $C_{m,q}$                   | -8.0446 | 1.1837  | -20.8571 |
| $C_{m,r}$                   | -0.0021 | -0.2081 | -2.4176  |
| $C_{m,\delta_{\mathtt{a}}}$ | 0.0177  | 0.9504  | 4.4178   |
| $C_{m,\delta_{e}}$          | -1.2524 | -0.0920 | 11.6916  |
| $C_{m,\delta_{ m r}}$       | 0.0165  | 0.0416  | 0.0795   |
| $C_{n,0}$                   | -0.0000 | -0.0000 | 0.0004   |
| $C_{n,b}$                   | 0.0682  | 0.0048  | -0.1193  |
| $C_{n,p}$                   | -0.0412 | -0.4284 | -1.0241  |
| $C_{n,q}$                   | -0.0007 | 0.0072  | 0.0489   |
| $C_{n,r}$                   | -0.0555 | 0.0316  | 0.1057   |
| $C_{n,\delta_{\mathrm{a}}}$ | 0.0234  | -0.0113 | -0.6566  |
| $C_{n,\delta_{e}}$          | -0.0000 | -0.0001 | 0.0014   |
| $C_{n,\delta_{ m r}}$       | -0.0509 | 0.0287  | -0.0572  |

**Table A2.** Aerodynamic moment stability derivatives.

## **B** AWE system parameters and constraints

The parameters and constraints of the wind profile and AWE system used in the simulations are summarized in Tables B1 and B2, respectively.

| Parameter                                                       | Value                                                         |  |  |
|-----------------------------------------------------------------|---------------------------------------------------------------|--|--|
| Logarithmic wind                                                |                                                               |  |  |
| Wind speed $u_{\text{ref}}  (\mathbf{m} \cdot \mathbf{s}^{-1})$ | 12                                                            |  |  |
| Reference height $z_{\text{ref}}$ (m)                           | 100                                                           |  |  |
| Roughness height $z_0$ (m)                                      | 0.0002                                                        |  |  |
| Aircraft                                                        |                                                               |  |  |
| Surface $S$ (m <sup>2</sup> )                                   | 150.45                                                        |  |  |
| Span $b$ (m)                                                    | 42.47                                                         |  |  |
| Chord $c$ (m)                                                   | 3.54                                                          |  |  |
| Mass $m_{\rm W}$ (kg)                                           | 6885.2                                                        |  |  |
|                                                                 | $5.768 \times 10^5$ 0 0                                       |  |  |
| Inertia tensor $\mathbf{J}$ (kg m <sup>2</sup> )                | $0 	 8.107 \times 10^4 	 0$                                   |  |  |
|                                                                 | $\begin{bmatrix} 0.47 & 0 & 6.5002 \times 10^5 \end{bmatrix}$ |  |  |
| Tether                                                          |                                                               |  |  |
| Drag coefficient $C_{\mathrm{D,T}}$ (-)                         | 1.2                                                           |  |  |
| Diameter $D_{\rm T}$ (m)                                        | 0.0297                                                        |  |  |
| Density $\rho_{\rm T}$ (kg m <sup>-3</sup> )                    | 971                                                           |  |  |

**Table B1.** Wind profile and AWE system parameters.

| Company in t                                                      | Path generation |          | Path tracking |          |
|-------------------------------------------------------------------|-----------------|----------|---------------|----------|
| Constraint                                                        | Min             | Max      | Min           | Max      |
| Aircraft operation                                                |                 |          |               |          |
| Cycle period $T$ (s)                                              | 0               | 20       | /             | /        |
| Position $\mathbf{q}[0], x$ (m)                                   | 0               | $\infty$ | 0             | $\infty$ |
| Position $\mathbf{q}[1], y$ (m)                                   | $-\infty$       | $\infty$ | $-\infty$     | $\infty$ |
| Position $\mathbf{q}[2], z$ (m)                                   | 2b              | $\infty$ | b             | $\infty$ |
| Rotation speed $\omega[0], p  (^{\circ} \mathrm{s}^{-1})$         | -10             | 10       | -50           | 50       |
| Rotation speed $\omega[1], q (^{\circ}s^{-1})$                    | -40             | 40       | -50           | 50       |
| Rotation speed $\omega[2], r (^{\circ}s^{-1})$                    | -25             | 25       | -50           | 50       |
| Aileron deflection $\delta[0], \delta_a$ (°)                      | -15             | 15       | -20           | 20       |
| Elevator deflection $\delta[1], \delta_{\rm e}$ (°)               | -7.5            | 7.5      | -10           | 10       |
| Rudder deflection $\delta[2], \delta_r$ (°)                       | -7.5            | 7.5      | -10           | 10       |
| Deflection rate $\dot{\delta}_{\rm a,e,r}  (^{\circ} \rm s^{-1})$ | -25             | 25       | -50           | 50       |
| Angle of attack $\alpha$ (°)                                      | -12             | 4        | -15           | 5        |
| Side-slip angle $\beta$ (°)                                       | -5              | 5        | -10           | 10       |
| Airspeed $V  (\mathrm{m \cdot s^{-1}})$                           | 10              | 120      | 10            | 120      |
| Acceleration (g)                                                  | -3              | 3        | -4            | 4        |
| Tether and winch                                                  |                 |          |               |          |
| Tether length $l$ (m)                                             | 10              | 1000     | 10            | 1000     |
| Tether acceleration $\ddot{l}  (\mathrm{m \cdot s^{-2}})$         | -2.5            | 2.5      | -5            | 5        |
| Tether force $F_{\rm T}$ (N)                                      | 50              | 1.7e6    | 50            | 1.7e6    |
| Power $P$ (MW)                                                    | -2.5            | 2.5      | -3            | 3        |

Power P (MW) -2 **Table B2.** AWE system constraints during path generation and tracking.

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
