# Peer review of "Aero-servo simulations of an airborne wind energy system using geometry-resolved computational fluid dynamics"

_Wind Energy Science, 2025_

## Referee Comment (RC1)

**Technical Review for WES-2025-73**

**Summary**:

This paper addresses the limitations of current airborne wind energy (AWE) simulations, which often rely on simplified aerodynamic models that cannot capture unsteady effects, flow separation, or the interaction between control surfaces and aircraft dynamics. To overcome this, the authors develop a geometry-resolved CFD framework—called the Virtual Wind Environment (VWE)—and couple it with the AWEbox simulation and control toolbox using an explicit aero-servo approach. They simulate a 1-loop power cycle of the MegAWES aircraft under realistic wind conditions and demonstrate accurate trajectory tracking and power output, achieving 96% of the reference performance. The study reveals aerodynamic effects missed by low-fidelity models and offers a high-fidelity tool for improving AWE design and control.

**nomenclature:**

It is better to add this section before the introduction to ease tracking the paper content.

**Introduction:**

Make this section right after adding a nomenclature section.

In the Introduction, the authors point out that most AWE simulations use simple aerodynamic models that miss unsteady flows and how moving control surfaces really work together. They give a fair overview of past studies, but don't show how big the errors usually are or explain in detail why earlier coupled methods fall short. They make a good case for using full-geometry CFD to see things like flow separation, but don't warn readers that this requires a lot more computing power. The term "Virtual Wind Environment" is a helpful name, but it isn't defined until after its first mentioned.
The authors use a linear aerodynamic model (AAM) for a fixed-wing aircraft, where the key aerodynamic effects are captured with matrix-based derivatives. They calculate those derivatives by running CFD simulations on an overset (Chimera) mesh, which is great for handling complex shapes and moving parts without having to remake the grid each time.

Still, the paper misses a few practical details. First, there's no information on how much computer time or memory these CFD runs need when you add more moving surfaces. Second, the authors never check their results against a known aircraft or wind-tunnel data, which would help prove their model really works under different flight conditions. Third, they don't show a mesh-independence study—refining the grid until the results

stop changing—to confirm their CFD setup is reliable. Because their model is linear, it may lose accuracy when control surfaces move a lot or during strong unsteady effects; discussing where that approach breaks down would round out the paper. The methods are solid, but adding notes on run-time, a basic validation case, a mesh-sensitivity check, and the limits of linearization would make the work much stronger.

**AWE system dynamics and control**

The presented work is clear that describes the six-degree-of-freedom dynamics, tether modeling, and MPC framework. To help readers fully understand and trust your approach, could you please elaborate on a few points? For instance, a brief explanation of how you selected the Baumgarte stabilization parameter and the MPC horizon length would be very insightful, as would a discussion of the controller's sensitivity to errors in the analytical aerodynamic model—have you tested how deviations in those coefficients affect tracking performance? It would also be helpful to know the computational effort required to solve the MPC every 5 ms (for example, whether it runs in real time on standard hardware or needs a high-performance cluster). Additionally, could you comment on the straight-tether assumption under strong crosswinds or slack conditions and whether you plan to extend the model to include tether sag or bending?  Since the tether's mass grows continuously as it reels out, some discussion of how that changing mass and inertia influence the dynamics and controller performance over a full power cycle would greatly clarify the robustness and applicability of your framework.

**Aero-servo coupling**

The aero-servo coupling you've implemented is very elegant in how it links the high-fidelity CFD solver with the AWEbox dynamics and MPC in a single explicit time-stepping loop, and using CoCoNuT to shuttle motion and force data keeps the workflow organized. To help readers appreciate and trust this approach, could you expand on a few aspects? For example, you mention that no coupling iterations are performed within each 5 ms timestep—have you observed any drift or stability issues over longer simulations, and how did you decide that single-step coupling was sufficient? It would also be useful to know the additional runtime cost and data-transfer overhead introduced by the CoCoNuT interface, especially when moving multiple overset zones each step. In the transition period where you blend AAM and VWE forces, what guided your choice of $n_1$ and $n_2$, and how sensitive are your results to that weighting schedule? Could you briefly describe any tests you ran to verify that the explicit coupling preserves energy consistency or avoids unphysical oscillations? A few sentences on these points would greatly strengthen confidence in the robustness and practicality of your aero-servo coupling.

**Results**

The Results section presents a clear demonstration of your coupled framework, showing both the trajectory tracking—where you achieve a close match within 4 m of the reference path—and the power output, with 96 % of the target average power captured. The side-by-side plots of aerodynamic forces and moments from the VWE and the analytical model effectively highlight model agreement and key discrepancies. To deepen the reader's understanding, it would be helpful to include quantitative error metrics (e.g., root-mean-square deviation over the cycle) for trajectory, power, and force comparisons

---

## Author Comment (AC1)

**Answer to reviewer 1**

**Technical Review for WES-2025-73**

***Summary***:

*This paper addresses the limitations of current airborne wind energy (AWE) simulations, which often rely on simplified aerodynamic models that cannot capture unsteady effects, flow separation, or the interaction between control surfaces and aircraft dynamics. To overcome this, the authors develop a geometry-resolved CFD framework— called the Virtual Wind Environment (VWE)—and couple it with the AWEbox simulation and control toolbox using an explicit aero-servo approach. They simulate a 1-loop power cycle of the MegAWES aircraft under realistic wind conditions and demonstrate accurate trajectory tracking and power output, achieving 96% of the reference performance. The study reveals aerodynamic effects missed by low-fidelity models and offers a high-fidelity tool for improving AWE design and control.*

**nomenclature:**

It is better to add this section before the introduction to ease tracking the paper content.

Thank you for this suggestion. However, we must follow the standard WES style without using nomenclature. Particular care is given to variable definition in the body of the manuscript.

**Introduction:**

Make this section right after adding a nomenclature section.

In the Introduction, the authors point out that most AWE simulations use simple aerodynamic models that miss unsteady flows and how moving control surfaces really work together. They give a fair overview of past studies, but don't show how big the errors usually are or explain in detail why earlier coupled methods fall short.

Thank you for that suggestion. However, it is difficult to quantify the error for the following reasons:

- These studies use different aircraft, making it difficult to compare aerodynamic data and power estimations. The error has not been quantified in previous studies so far.
- The additional complexity of the crosswind flight maneuver, with all its parameters, makes it additionally difficult to perform a direct comparison.
- Not a lot of wind tunnel or experimental data is available for airborne wind energy systems, and also not for the reference aircraft that is studied in this work.

However, by opting for this reference aircraft, the results can be made available to the research community for later comparison.

A study has been performed by a colleague comparing the prediction of the lift curve with AVL, which uses the vortex lattice method, and CFD (see figure below). For an angle of attack of 5 degrees, the lift predicted by CFD is **25%** less. This deviation is explained by the prediction of the stall by the CFD. The prediction of AVL is in line with the original prediction of [Eijkelhof et al.].

[Figure]

Furthermore, in "Heydarnia, O., Wauters, J., Lefebvre, T., & Crevecoeur, G. (n.d.). *Opti-MegAWES: A toolbox for optimal path planning of megawatt-scale airborne wind energy systems.*", A preliminary study is performed comparing the resulting power corresponding to these aerodynamic predictions. This study reveals a **30%** reduction in power for a wind speed of 20m/s when comparing results from a VLM (AVL) prediction to a CFD prediction.

They make a good case for using full-geometry CFD to see things like flow separation, but **don't warn readers that this requires a lot more computing power**.

Indeed, the computational cost is now highlighted in the conclusion on line 470 c1:

*(c1) However, it should be noted that full-geometry CFD simulations are significantly more computationally demanding, rendering them impractical for rapid design iterations.*

*(c1) Despite its capabilities, the current framework also exhibits certain limitations. First, it should be noted that full-geometry CFD simulations are significantly more computationally demanding, rendering them impractical for rapid design iterations.*

Furthermore, a dedicated section on computational time is added at the end of the results section.

**6.4 Notes on computational time**

*c1 The simulation was performed on a system with 2 × 20-core Intel Xeon Gold 6242R processors (3.1 GHz) and 187.4 GB of system memory. The peak memory usage during the simulation reached 92.2 GB. A breakdown of the computational time for one power loop cycle is provided in Table 1. Furthermore, it is calculated that the 63% of the Fluent solver calculation time is attributed to the overset technique. It can also be observed that a substantial portion of the total simulation time was spent on data saving, indicating a potential area for optimization in future runs.*

*c2 The model predictive control (MPC) algorithm required approximately 0.6 s of wall time per time step. While this is relatively efficient in the context of the full simulation, it exceeds the simulation time step of 5 ms, making this control configuration unsuitable for real-time implementation. Note that the MPC calculation time depends on implementation and hardware. Faster implementations are available, such as acados (Verschueren et al., 2020).*

The term "Virtual Wind Environment" is a helpful name, but it isn't defined until after its first mentioned.

This term is defined in the abstract (line 9), and at its first use in the main text, so in the introduction (line 62).
The authors use a linear aerodynamic model (AAM) for a fixed-wing aircraft, where the key aerodynamic effects are captured with matrix-based derivatives. They calculate those derivatives by running CFD simulations on an overset (Chimera) mesh, which is great for handling complex shapes and moving parts without having to remake the grid each time.

Still, the paper misses a few practical details.

First, there's no **information on how much computer time or memory these CFD runs need when you add more moving surfaces**.

A section dedicated to computer time and memory use is added to the result section.

Furthermore, a small study is performed to assess the additional time of the control surfaces and overset. The study was executed using a smaller background of around 300,000 cells using a simulation run from 10 to 20 timesteps of 0.005s (timesteps 0-10 are used for initialization) of a horizontal flight at 80 m/s for the following configuration: wing only (no overset), wing + background, wing + ailerons + background, wing + ailerons + horizontal tail (HT) + background, full aircraft + background. This study reveals that the computational time per iteration over the total number of cells (slope in the figure below) increases with adding overset components, but the increase decreases with the number of components added. For the simulation of wing + background (**1** overset component),

the ratio of computational time over the total number of cells increases by **120 %,** compared to the simulation without overset (wing only). For the simulation of the full aircraft + background (**6** overset components), the increase in ratio of computational time over the total number of cells increases by **168%,** compared to the simulation without overset (wing only). In conclusion, **63%** (168/(100+168)) of the total computational time is attributed to the overset connectivity. This conclusion is added to the section on computational time.

[Figure]

No significant changes in memory use have been observed for the different configurations. The memory required for the mesh and overset is relatively small compared to the memory needed to load the executables in the parallel simulation. Therefore, the slight increase related to the overset cannot be accurately determined with this setup. For reference, a large job allocates around 50 GB for all processors, while the overset grid accounts for only ~5 GB of that.

Second, the authors **never check their results against a known aircraft or wind-tunnel** data, which would help prove their model really works under different flight conditions.

Not a lot of wind tunnel or experimental data is available for airborne wind energy systems in general, and certainly not for the reference aircraft that is studied in this work as it has never been built. See previous comment for comparison with different simulations. In future work, a validation campaign is planned for existing aircraft (AP2/3).

Third, they don't show **a mesh-independence study**—refining the grid until the results stop changing—to confirm their CFD setup is reliable.

A mesh-independence study is carried out in previous work: Pynaert, N., Haas, T., Wauters, J., Crevecoeur, G., and Degroote, J.: Wing deformation of an airborne wind energy system in crosswind flight using high-fidelity fluid-structure interaction, Energies, 16(2), https://doi.org/10.3390/en16020602, 2023:

*Mesh independence study (copied from Pynaert et al, 2023): Four meshes are constructed for this study based on the finest possible mesh in the range of validity of wall functions. The finest mesh is indicated as ultra-fine in Table 2. A constant refinement factor of (3/2)^(1/2) is applied among all levels (ultra-fine, fine, medium, and coarse) and is applied to the number of divisions. Mesh convergence can be observed in Figure 10. The lift coefficient changes only by 0.7%, and the drag coefficient changes by 4% in the last refinement, while the number of cells is almost doubled. The study shows that using the fine grid is a good compromise between accuracy and computational cost.*

**Table 2.** Description of meshes used for mesh independence study.

|  | Number of Cells |
| --- | --- |
| Coarse | $0.76 \times 10^6$ |
| Medium | $1.40 \times 10^6$ |
| Fine | $2.56 \times 10^6$ |
| Ultra-fine | $4.76 \times 10^6$ |

[Figure]

**Figure 10.** Lift and drag coefficient versus the number of cells in the component mesh.

The mesh used in this study for the wing resembles the **ultra-fine** mesh of the previous study. A reference to this mesh refinement study has been added in the manuscript on line 116:

*C3 The grid of the wing is based on the grid refinement study that was performed in (Pynaert et al., 2023).*

Because their model is **linear**, it may lose accuracy when control surfaces move a lot or during strong unsteady effects; discussing where that approach breaks down would round out the paper.

It should be noted that both the introduced models, e.g. the analytical aerodynamic model (AAM) based on aerodynamic derivatives (calculated here with CFD), and the virtual wind environment (VWE), fully based on CFD, are non-linear. However, the difference between the models is that the AAM is quasi-static and the VWE is time-dependent. A comparison between both models is provided in the results, see section 6.2.

The methods are solid, but adding **notes on run-time**, **a basic validation case**, **a mesh-sensitivity check**, and the **limits of linearization** would make the work much stronger.

**AWE system dynamics and control**

The presented work is clear that describes the six-degree-of-freedom dynamics, tether modeling, and MPC framework. To help readers fully understand and trust your approach, could you please elaborate on a few points?

For instance, a brief explanation of how you **selected the Baumgarte stabilization parameter** (1) **and the MPC horizon length** (2) would be very insightful, as would a discussion of the **controller's sensitivity to errors** in the analytical aerodynamic model—have you tested how deviations in those coefficients affect tracking performance? (3)

1. These are the default values of AWEbox that were selected in previous work [De Schutter et al.]. A detailed discussion about Baumgarte stabilization is out of the scope of the paper, and it is chosen to leave this out. The reader is redirected to the original paper of the AWEbox methods for these technical details on line 276:

*For a detailed description of the solution methods for the optimal control formulations, the reader is referred to De Schutter et al. (2023).*

2. The current settings were established empirically to ensure simulation stability. This is now clarified on line 279:
*c1 The current settings were established empirically to ensure simulation stability.*
Future work could aim to develop a systematically optimized and practically implementable controller.

3. A systematic sensitivity study on the controller is not performed and is not the focus of this paper. However, during the development, the controller proved quite sensitive to the aerodynamic derivatives, and that's why CFD-based aerodynamic derivatives have been derived to ensure minimal model mismatch. Future studies could be directed towards the analysis of the controller's sensitivity to the model.

It would also be helpful to know the **computational effort** required to solve the MPC every 5 ms (for example, whether it runs in real time on standard hardware or needs a high-performance cluster).

This is added in the additional section on computational effort, line 444:

*c2 The model predictive control (MPC) algorithm required approximately 0.6s of wall time per time step. While this is relatively fast in the context of the full simulation, it exceeds the simulation time step of 5ms, making this control configuration and its*

*implementation unsuitable for real-time implementation. Note that the MPC calculation time depends on implementation and hardware. Faster implementations are available, such as acados (Verschueren et al., 2020).*

Additionally, could you comment on the **straight-tether assumption** under strong crosswinds or slack conditions and whether you plan to extend the model to include tether sag or bending?

The straight-tether assumption is now further commented on line 205 c1:

*c1 In Malz et al. (2019), it was found that the straight tether assumption is adequate for estimating power generation in a small-scale airborne wind energy (AWE) system (specifically, the AP2 developed by the former Ampyx Power). In contrast, the study of \citep{Heydarnia}, based on the MegAWES aircraft, concluded that the straight tether assumption can lead to the overestimation of harvested power up to 33%. Future work will focus on incorporating tether sag into the system dynamics model.*

Since the tether's mass grows continuously as it reels out, some discussion of how that **changing mass and inertia influence the dynamics** and controller performance over a full power cycle would greatly clarify the robustness and applicability of your framework.

These effects (changing mass and inertia) are included in the simulation, but a detailed analysis of tether effects is out of the scope of this report, as the focus is on the aircraft aerodynamics. For this analysis, you can refer to "Heydarnia, O., Wauters, J., Lefebvre, T., & Crevecoeur, G. (n.d.). *Opti-MegAWES: A toolbox for optimal path planning of megawatt-scale airborne wind energy systems*."

**Aero-servo coupling**

*The aero-servo coupling you've implemented is very elegant in how it links the high-fidelity CFD solver with the AWEbox dynamics and MPC in a single explicit time-stepping loop, and using CoCoNuT to shuttle motion and force data keeps the workflow organized.*

*To help readers appreciate and trust this approach, could you expand on a few aspects?*

For example, you mention that **no coupling iterations** are performed within each 5 ms timestep—have you observed any drift or stability issues over longer simulations, and how did you decide that single-step coupling was sufficient?

Currently, 3 loops have been flown with this framework (not part of the paper), and no drift or stability issues have been encountered. So we can conclude that the single-step coupling is sufficient, also for future simulations.

It would also be useful to know **the additional runtime cost and data-transfer** overhead introduced by the CoCoNuT interface, especially when moving multiple overset zones each step.

The additional runtime cost and data transfer are now presented in the section on computational time: Time for "Coupling" and "Save-time" represents this contribution.

*c1 The simulation was performed on a system with 2 ×20-core Intel Xeon Gold 6242R processors (3.1 GHz) and 187.4GB of system memory. The peak memory usage during the simulation reached 92.2GB. A breakdown of the computational time for one power loop cycle is provided in Table 1. Notably, a substantial portion of the total simulation time was spent on data saving, indicating a potential area for optimization in future runs.*

The additional runtime for moving multiple overset zones is discussed in one of the previous comments. This runtime is part of the "Fluent solver".

**Table 1.** Computational time of a 1-loop power cycle.

| Component | # of Cores | Hours | Per time step (s) |
|---|---|---|---|
| Total run time (4000 time steps) | | 174.6 | 157.1 |
| Fluent solver | 40 | 130.4 | 117.4 |
| Coupling | 1 | 2.9 | 2.6 |
| Save time (coupling) | 1 | 41.3 | 37.2 |
| MPC wall time | 1 | 0.7 | 0.6 |

In the transition period where you blend AAM and VWE forces, what guided **your choice of $n_1$ and $n_2$**, and how sensitive are your results to that weighting schedule? Could you briefly describe any tests you ran to verify that the explicit coupling preserves energy consistency or avoids unphysical oscillations?

The choice of n1 (220 = 1.1s) and n2 (440 = 2.2s) is based on the Wagner functions (see plot below) calculated for the initial conditions of the aircraft: after n1, ~90% of the lift is reached, and after n2, ~95% of the lift is reached. By assuming these values sufficient margin is allowed for the flow to develop, and not a lot of sensitivity is expected. The physical explanation is added to the text on line 353 c3:

*c3 This timestep corresponds to 2.2s, after which the starting vortex is sufficiently distant and startup effects are considered negligible.*

No unphysical oscillation has been observed in the results. The weighting function is sufficiently smooth.

[Figure]

Source: https://flow.byu.edu/Aeroelasticity.jl/dev/models/aerodynamics/wagner/

A few sentences on these points would greatly strengthen confidence in the robustness and practicality of your aero-servo coupling.

**Results**

The Results section presents a clear demonstration of your coupled framework, showing both the trajectory tracking—where you achieve a close match within 4 m of the reference path—and the power output, with 96 % of the target average power captured.

The side-by-side plots of aerodynamic forces and moments from the VWE and the analytical model effectively highlight model agreement and key discrepancies. To deepen the reader's understanding, it would be helpful to **include quantitative error metrics** (e.g., root-mean-square deviation over the cycle) for trajectory, power, and force comparisons.

Thank you for this valuable suggestion; these metrics have been added to the result section on line 379 c1:

… *c1 and a root mean square deviation of 1.6m over the cycle.*

, line 381 c3:

*c3 The root mean square deviation over the cycle amounts 93kW and the reduction in average power is 16kW, which represents 4% of the reference average power.*

, line 393 c3:

*c3 A comparison is now made between the resulting forces obtained from the VWE and those predicted by the AAM, the model employed within the MPC framework. The resulting forces from the AAM and VWE exhibit a consistent trend. The maximum force deviations are 0.080 for Cz, 0.007 for Cy, and 0.008 for Cx, **and the root mean square deviation over the cycle is 0.025 for Cz, 0.002 for Cy, and 0.004 for Cx**. The CFD framework used to derive the stability derivatives for the AAM also forms the foundation*

*of the VWE. Therefore, the remaining discrepancies likely arise from the AAM's limitations in capturing nonlinear aerodynamic effects beyond the angle of attack, as well as unsteady aerodynamic phenomena.*

, and line 406 c1:

*c1 The maximum offset between the two models amounts to 0.01 for the roll Cl, 0.03 for the pitch Cm, and 0.004 for the yaw Cn moment coefficient, respectively. **The root mean square deviation between the two models over the cycle amounts to 0.006 for the roll Cl, 0.01 for the pitch Cm, and 0.001 for the yaw Cn moment coefficient, respectively.***

**Answer to reviewer 2**

Thanks for your hard work and for writing this up for the world to read! I have summarized my remarks per section and most of the style points separately. I was wondering if it is correct that you are making **no data or any code available**?

The code will become available after acceptance in a GitHub repository, and the data in a repository on Zenodo.

**Style Points**

**-** Descriptive subscripts should not be italic, and should be in straight font instead. (see #7 https://physics.nist.gov/cuu/Units/checklist.html: "Superscripts and subscripts are in italic type if they represent variables, quantities, or running numbers. They are in roman type if they are descriptive.") Done.

-  Minimize the number of brackets, e.g. Figure 5 caption could be "…(a) xz-plane at y = 1.3m and in the (c).." or in line 322.  Done.

- WES style guide mandates that you use acronyms for Eqs. and Fig. etc. except when at the beginning of the sentence, see the WES style guide and adjust accordingly. (eq. line 322) Done.

- You should include all numbers in mathmode, now you are including some and others not, which is not consistent and is not following style guides. Done.

- Units should be (m) and not [m] Done.

- WES demands colorblind readable plots, yours are not. (e.g. Fig 12) Done.

- Units should be m s^-1 and not m/s, e.g. line 383 deg/s Done.

- You could move author contributions, competing interests and acknowledgements in front of the appendices Done.

Thank you for addressing these style points; they have been addressed in the revised version.

**Abstract**

"To provide meaningful insights into crosswind flight maneuvers they must incorporate.." I would write instead, should incorporate as it is not the case that one can not get meaningful insights at all without incorporating the additional effects. It is just that the insights get more accurate/more useful. Ok, adapted on line 5 c1.

As you can never be 100% certain that your work is a first of its kind, it's better to use the word "novel" in the abstract. Ok, adapted on line 12 and line 72 .

**Introduction**

Paragraph on lines 34--46 or so, there you could add a reference to D. Eijkelhof's work, as you are discussing fixed-wing simulation works. Ok, added on line 36 c1:
*c1 Eijkelhof et al. (2023) developed an aerodynamic toolchain for the design analysis of AWE systems with a box-shaped wing using steady Reynolds-averaged Navier–Stokes (RANS) simulations. Vimalakanthan et al. (2018) conducted RANS simulations of an aircraft that included control surfaces. However, both studies focus on a steady horizontal flight, neglecting crosswind motion and setting the control surfaces in a fixed position.*

You also don't discuss in the introduction that you are neglecting deformation, why you are doing it, why that is okay, and what the effect is. This discussion is added to the conclusion on line 477:

*c3 Secondly, the wing deformation and fluid -structure interaction (FSI) effects are neglected in this study, based on the findings of Pynaert et al. (2023), which showed that structural deformations of the investigated aircraft were minimal. Specifically, the FSI analysis indicated only a 1.4% reduction in aerodynamic loads and a wing tip deflection of at most 0.3 m. Nevertheless, the framework has been designed to support the inclusion of FSI in future studies involving more flexible aircraft configurations.*

Sentence 51: "Specifically, they fail to account for unsteady aerodynamic phenomena and omit the interaction of various aerodynamic effects - such as the influence of the rotational speed on aileron effectiveness." -- rephrase, into something like: "they fail to account for various aerodynamic effects, including unsteady and .." Ok, adapted on line 54.

*C5 Specifically, they fail to account for various aerodynamic effects, including unsteady phenomena and interactions, such as the influence of the rotational speed on the effectiveness of the ailerons. c6 Such effects can lead to violations of the constraints defined during trajectory optimization, potentially resulting in degraded performance or even structural failure. Given the high cost of flight testing, accurate CFD tools are*

*essential to predict and mitigate these effects during the design and planning stages.*

**2. Virtual Wind Environment**

line 100--105: Explain briefly what the overset technique is and how it works, could be 1 sentence. Ok, added on line 110 c1:

*c1 This overset connectivity allows for interpolation of the flow variables between the separate grids as explained in Sect. 3.4.*

Earlier, you talked about resolving local flow phenomena accurately, and in line 110, you state using a **y+ value of 100**. This means that boundary layer properties (which are local flow phenomena) are not captured accurately. This is not to say that y+ of 100 you can't meaningfully resolve any local flow phenomena, but then you should be more open and clear about the limitations, and what you are resolving and what not. Also, at current, there is no mention of why y+ is 100, would be good or bad, or why it is chosen.

It was clarified that wall functions are valid for y+ between 30-500 on line 113. Additionally, it is now stated that wall functions are used to limit the computational expense and that they limit the accuracy in predicting flow separation on line 113:

*c2 While this approach reduces the accuracy in prediction flow separation, it provides a reasonable overall impression of the flow around the aircraft and limits the computational expense of the simulation.*

Additionally, this limitation has been added to the conclusion on line 472:

*c2 To limit the computational expense of the simulations presented in this work, the boundary layer at the aircraft surface is not fully resolved; instead, wall functions are employed. While this approach reduces the accuracy in predicting flow separation, it provides a reasonable overall impression of the flow around the aircraft. Nevertheless, the current framework is compatible with further mesh refinement.*

You mention the $z_0$ value twice (line 139). Ok, corrected at line 149 c1.

Line 145, if this is in the local frame, please add this. Ok, added at line 155 c1 and caption Figure 5 c2-6.

**4. AWE system dynamics**

Skew operator is in the wrong order, line 190. The equation has been removed to simplify the discussion on the dynamics.

Add some space between Fig. 6 and eq. 5 and 6. Done.

Why don't you use the terminology kite rather than aircraft? Kite is defined as anything from the bridle point up, which in your configuration is identical to the aircraft itself.

Thank you for this suggestion. We opted for aircraft as a semantic choice to not confuse less acquainted readers with soft kites.

You talk about 3D forces, then you should use capital subscripts: C_L, C_M, C_N, etc. Subscripts l, m, and n are used to indicate the roll, pitch, and yaw moment, respectively.

Line 257 could leave the "of 1" out. The sentence is smoother without. Ok, corrected on line 268 c1.

**5. Aero-servo coupling**

An itemized list in lines 275--282, rather than having the steps described in a line, would be much easier to follow as a reader. Done.

Equations around line 325, you can't add text to the equations like you are doing there. Ok, this is corrected at line 340 c1,c2.

**6. Results**

Fig 10. Use LaTeX font, or even generated labels, and make them non-overlapping. Done.

Line 365, rewrite, the use of - here is not so clear. Rewritten at line 383 c1:

*c1 The aerodynamic properties of this coupled simulation (COSIM), more specifically the angle of attack $\alpha$ (a), the side-slip $\beta$ (b), and the apparent wind speed Va (c), are plotted in the left column of Fig. 12.*

Line 369, could remove VWE from brackets and add in text. Corrected at line 387 c2.

Line 373 rewrite. See next comment.

Line 365- 378: You leave the reader with some questions here. Make it more explicit that the 'minimal model mismatch' is only visible here, and later, the differences are clearer. From this part of the analysis, one thinks from the intro that large differences will be

seen, and then suddenly argues that one anticipates small differences, which is strange. This paragraph has been rewritten on line 391 c3:

*c3 A comparison is now made between the resulting forces obtained from the VWE and those predicted by the AAM, the model employed within the MPC framework. The resulting forces from the AAM and VWE exhibit a consistent trend. The maximum force deviations are 0.080 for Cz, 0.007 for Cy, and 0.008 for Cx, and the root mean square deviation over the cycle is 0.025 for Cz, 0.002 for Cy, and 0.004 for Cx. The CFD framework used to derive the stability derivatives for the AAM also forms the foundation of the VWE. Therefore, the remaining discrepancies likely arise from the AAM's limitations in capturing nonlinear aerodynamic effects beyond the angle of attack, as well as unsteady aerodynamic phenomena.*

Figure 12: Why is blue not shown on the left? What is the consequence of **the LARGE deviation in sideslip**?

The plot's color scheme has been improved for clarity. As for the sideslip behavior:

- The sideslip angle is not actively tracked, so deviations from the reference are to be expected.

- The constraints in the optimal control problem for trajectory generation and tracking are different; the ones for tracking are more relaxed. The system optimization tends to drive the sideslip angle toward its limits (±10°), which is consistent with the reference trajectory (although it is limited to ±5° during the trajectory generation). This is now also clarified in the paper on line 400:

  *c1 Note that this constraint for trajectory generation is ±5◦.*

- This behavior is likely a compensation mechanism for the reduced aileron effectiveness in the VWE scenario.

- It also helps to counteract yawing moments, contributing to coordinated flight under control limitations.

Potential consequence:
A possible drawback is a reduction in aerodynamic efficiency due to a lower lift-to-drag ratio (L/D). However, this effect has not been observed directly in the simulations.
A more detailed analysis would be needed to isolate the impact of sideslip and confirm this hypothesis, which lies beyond the scope of this paper.

Figure 13 (b): Why is there this difference? And can you make the legend external? The overlap with lines is not so nice.

These plots have been improved in this version.

The largest difference in the elevator deflection with the reference trajectory, and as a result also in the pitch rate q at 8.4s, corresponds to a steering input/correction at the beginning of the transition between the power generation and consumption phase.

Line 388, do you mean $C\_M$ instead of $C\_l$? This discussion relates to the roll moment coefficient $C\_l$.

Line 418, you could combine sentences and leave out "finally".

Line 439 *c1 The wall shear stress in the xB-direction along the aircraft surfaces is examined, where negative values indicate flow reversal and signal regions of flow separation.*

**7. Conclusion and outlook**

Line 430, mention which effects are not captured and which deviations are found.

The most important contribution is repeated at line 461 c4:

*... c4, such as the nonlinear contribution of the rotational motion on the control surfaces' effectiveness and force/moment coefficients.*

**Appendices**

Formatting of equations A4, A5, A6. should be improved. Done.

Avoid double subscripts at all costs, could use, to separate, e.g. ,line 481 $C\_i,j$. Done.